# Unique ligand and kinase-independent roles of the insulin receptor in regulation of cell cycle, senescence and apoptosis

Hirofumi Nagao[1], Ashok Kumar Jayavelu[2,3], Weikang Cai[1,4], Hui Pan [5], Jonathan M. Dreyfuss [5], Thiago M. Batista[1], Bruna B. Brandão [1], Matthias Mann [2] & C. Ronald Kahn [1]✉

Insulin acts through the insulin receptor (IR) tyrosine kinase to exert its classical metabolic and mitogenic actions. Here, using receptors with either short or long deletion of the β-subunit or mutation of the kinase active site (K1030R), we have uncovered a second, previously unrecognized IR signaling pathway that is intracellular domain-dependent, but ligand and tyrosine kinase-independent (LYK-I). These LYK-I actions of the IR are linked to changes in phosphorylation of a network of proteins involved in the regulation of extracellular matrix organization, cell cycle, ATM signaling and cellular senescence; and result in upregulation of expression of multiple extracellular matrix-related genes and proteins, down-regulation of immune/interferon-related genes and proteins, and increased sensitivity to apoptosis. Thus, in addition to classical ligand and tyrosine kinase-dependent (LYK-D) signaling, the IR regulates a second, ligand and tyrosine kinase-independent (LYK-I) pathway, which regulates the cellular machinery involved in senescence, matrix interaction and response to extrinsic challenges.

Insulin is produced by pancreatic β-cells and acts as a major anabolic hormone in the control of glucose, lipid, and protein homeostasis[1,2]. At the cellular level, these effects are mediated by the insulin receptor (IR) and its intrinsic tyrosine kinase activity, which is activated by ligand binding. This initiates a cascade of phosphorylation events that lead to the activation or inhibition of multiple intracellular enzymes, changes in nutrient uptake, and regulation of gene expression for many proteins involved in the control of metabolism and growth[2,3]. These occur through two major canonical signaling post-receptor pathways: the IRS-1/phosphatidylinositol 3-kinase (PI3K)/Akt pathway, which is linked to most metabolic actions of insulin, and the Shc/Ras/MAP kinase pathway, which is linked to regulation of cellular and organismal growth and differentiation[4,5]. Perturbations of these pathways in the

insulin-resistant state are central to the pathogenesis of type 2 diabetes, metabolic syndrome, and many other disorders[3,6–8]. Many of these actions are shared by the closely related IGF-1 receptor (IGF1R), which preferentially binds IGF-1 and -2 but has a highly homologous receptor kinase and an overlapping signaling cascade[9,10]. In general, however, IR preferentially stimulates the phosphorylation of proteins associated with PI 3-Kinase, Akt, and mTORC1 pathways, whereas IGF1R preferentially stimulates the phosphorylation of proteins associated with cell cycle, mitosis pathways, and the Rho-GTPase pathway[9,10]. At least part of this differential signaling is due to differences in the sequence within the juxtamembrane domain of these receptors, which for IR includes the sequence NPEY followed by leucine, which favors the recruitment of the receptor substrate IRS-1,

[1]Section of Integrative Physiology and Metabolism, Joslin Diabetes Center, Harvard Medical School, Boston, MA 02215, USA. [2]Department of Proteomics and Signal Transduction, Max Planck Institute of Biochemistry, 82152 Martinsried, Germany. [3]Proteomics and Cancer Cell Signaling Group, Clinical Cooperation Unit Pediatric Leukemia, German Cancer Research Center (DKFZ), Heidelberg, Germany. [4]Department of Biomedical Sciences, New York Institute of Technology College of Osteopathic Medicine, Old Westbury, NY 11568, USA. [5]Bioinformatics and Biostatistics Core, Joslin Diabetes Center, Harvard Medical School, Boston, MA 02215, USA. ✉e-mail: c.ronald.kahn@joslin.harvard.edu

whereas, in the IGF1R, the NPEY is followed by phenylalanine, which favors recruitment of Shc as a substrate[9]. In addition, sequences in the C-terminus of IR that contain two sites of tyrosine phosphorylation have been suggested to serve as an inhibitory regulator of mitogenesis, such that removal or mutation of the C-terminus has been reported to enhance growth signaling by the IR[11,12].

In addition to these ligand-activated and tyrosine kinase-dependent events, we have presented data suggesting that some biological effects mediated by the IR may be tyrosine kinase-independent. For example, cells lacking both IR and IGF1R (DKO cells) are resistant to apoptosis[13], and this effect is at least partially rescued by the re-expression of either wild-type IR or a kinase-dead mutant of IR (K1030R)[13]. DKO cells also show changes in DNA methylation and altered expression of multiple imprinted genes and miRNAs[14], however, the full range of these effects, how they are mediated, and their relationship to receptor structure are unknown. Effects from other types of receptors in the unoccupied or unliganded state have also been reported. For example, for many G-protein coupled receptors, inhibitory signals arise from the unoccupied receptor via interactions with arrestin proteins, and these signals can be diminished by agonist or antagonist binding[15]. Evidence has also been presented for the regulation of osteoclast apoptosis by the unoccupied αvβ3 integrin[16] and the regulation of neuronal apoptosis by the unliganded neurotrophin receptor[17]. The mechanisms underlying these actions, however, remain unknown.

To define the receptor-dependent but ligand- and kinase-independent signaling for IR, we have generated preadipocytes in which the endogenous IR and IGF1R have been genetically inactivated to create DKO cells. These cells have then been reconstituted with the normal wild-type IR, a kinase-dead mutant of IR (K1030R)[13], a truncated IR lacking 79 amino acids from the C-terminus (ΔCT), or a truncated IR lacking almost the entire intracellular domain including both tyrosine-kinase domain and C-terminus, leaving only the intracellular juxtamembrane 36 amino acids (Juxtamembrane-Domain-Only, JMO), after which we performed comprehensive molecular phenotyping, including global phosphoproteomics, gene expression, and protein expression analysis, to understand the domain-dependent roles of the occupied and unoccupied IR.

Here we show that, in addition to the classical ligand and tyrosine kinase-dependent (LYK-D) signaling, the IR, even in the unliganded state, modulates a network of protein phosphorylation events linked to the control of expression of genes and proteins in several key regulator pathways. These events are largely dependent on the presence of the intracellular domain of the IR, but independent of receptor kinase activity, the receptor C-terminus or receptor-ligand binding. These effects include a major remodeling of the extracellular matrix (ECM), expression of cell cycle and immune-related genes and proteins, and a senescence-associated phenotype. This ligand- and tyrosine kinase-independent (LYK-I) signals also regulate sensitivity to apoptosis by both intrinsic and extrinsic pathways. Together these studies indicate that a second, previously unrecognized signaling state of the IR is independent of ligand occupancy or kinase activity, but plays an important role in the biology of the cell.

## Results

### Kinase- and domain-dependent roles of IR in regulating metabolism and growth

To identify the kinase- and domain-dependent roles of IR, we generated preadipocytes in which both endogenous IR and IGF1R had been genetically inactivated using Cre-lox recombination[13], and then reconstituted these DKO cells with the full-length human IR (B isoform), the hIR with a K1030R mutation (kinase dead), a truncated hIR lacking the 79 amino acids of the C-terminus, which contains two sites of tyrosine phosphorylation (ΔCT), and a truncated hIR with only the 36 juxtamembrane intracellular amino acids, thus lacking both the

tyrosine-kinase domain and the C-terminus (JMO) (Fig. 1a). All the exogenous receptor constructs were FLAG-tagged. As expected, DKO cells showed no expression of the IR or IGF1R at either the mRNA or protein level (Fig. 1b, c and Supplementary Fig. 1a, b), while cells reconstituted with intact IR or K1030R, ΔCT, and JMO mutants had similar mRNA expression of these recombinant receptors (Fig. 1b). On average, the mRNA levels of the exogenously expressed receptor constructs were about four times higher than that of the endogenous IR in wild-type (WT) cells, indicating only a modest degree of over-expression. The full-length IR and mutated IRs were also similarly expressed at the protein level and predominately localized to the membrane fraction by immunoblotting, with β-subunits of the predicted length for each mutant (Fig. 1c). A small fraction of each of these receptors could also be identified in extracts of the nuclear fraction, and this did not change with 30 min of insulin stimulation (Supplementary Fig. 1c). In cells expressing the normal IR and ΔCT mutant, insulin stimulation led to robust levels of receptor autophosphorylation, which was accompanied by increased phosphorylation of IRS-1[Y608] and Akt[S473] (Fig. 1d). As expected, stimulation of the K1030R or JMO mutants lead to no autophosphorylation, IRS or Akt phosphorylation, consistent with their lack of tyrosine kinase activity. Recruitment of IRS-1 to the IR was induced by insulin in IR and ΔCT cells, as judged by co-precipitation, but was absent in cells lacking a functional kinase domain, i.e., K1030R and JMO cells (Supplementary Fig. 1d). Consistent with their signaling activity, using a standard differentiation cocktail[18], only cells expressing the intact IR and ΔCT mutants became differentiated and contained increased amounts of lipid droplets (Fig. 1e and Supplementary Fig. 1e). Preadipocytes expressing IR and ΔCT also showed the highest rates of cell proliferation, followed by cells expressing the IR K1030R mutant cells, with lowest proliferation in DKO and JMO cells (Fig. 1f). Likewise, mRNA levels of phosphofructokinase liver type (*Pfkl*) were significantly upregulated by 6 h of insulin stimulation only in cells expressing intact IR and the ΔCT mutant (Fig. 1g). Cells expressing the IR mutations also showed differential effects on glycolysis with insulin stimulation significantly increasing glycolysis as measured in a Seahorse Metabolic Analyzer in preadipocytes expressing IR and ΔCT, but decreasing glycolysis in the cells expressing IR-JMO (Fig. 1h). Together, these data demonstrate that the intracellular domain of the IR, including a functional kinase domain but not the C-terminal 79 amino acids, is critical for ligand-stimulated glycolysis, regulation of *Pfkl*, and adipocyte differentiation. Cells expressing the K1030R mutant also showed significantly higher cell proliferation than DKO cells, but not to the extent of cells with a kinase-active receptor. Cells expressing IR-JMO also may have some unique effects, as illustrated by the effect to reduce glycolysis.

### Differential phosphoproteomic signature of DKO, IR, K1030R, ΔCT, and JMO cells

To define differences in signaling pathways regulated in DKO, IR, K1030R, ΔCT, and JMO cells, confluent cells expressing these receptors (Supplementary Fig. 2a) were serum starved for 6 h and stimulated with 100 nM insulin for 15 min or left untreated, and after which proteins were extracted and subjected to a global phosphoproteomic analysis by liquid chromatography–mass spectrometry (LC–MS/MS)[19] (Supplementary Data 1). As described in more detail in Methods, we included in the analysis phosphosites that were detected in at least 50% of all samples, including basal and DKO samples, so that we would not lose phosphosites only present cells with IR alterations that affect the classical signaling pathways. It is particularly important to retain such phosphosites in data-dependent acquisition (DDA) phosphoproteomics, as missing values often indicate the absence of phosphorylation rather than being missing completely at random. Of the 14,971 phosphopeptides kept for analysis, values were considered undetectable in only 21% of all samples under either the basal or stimulated

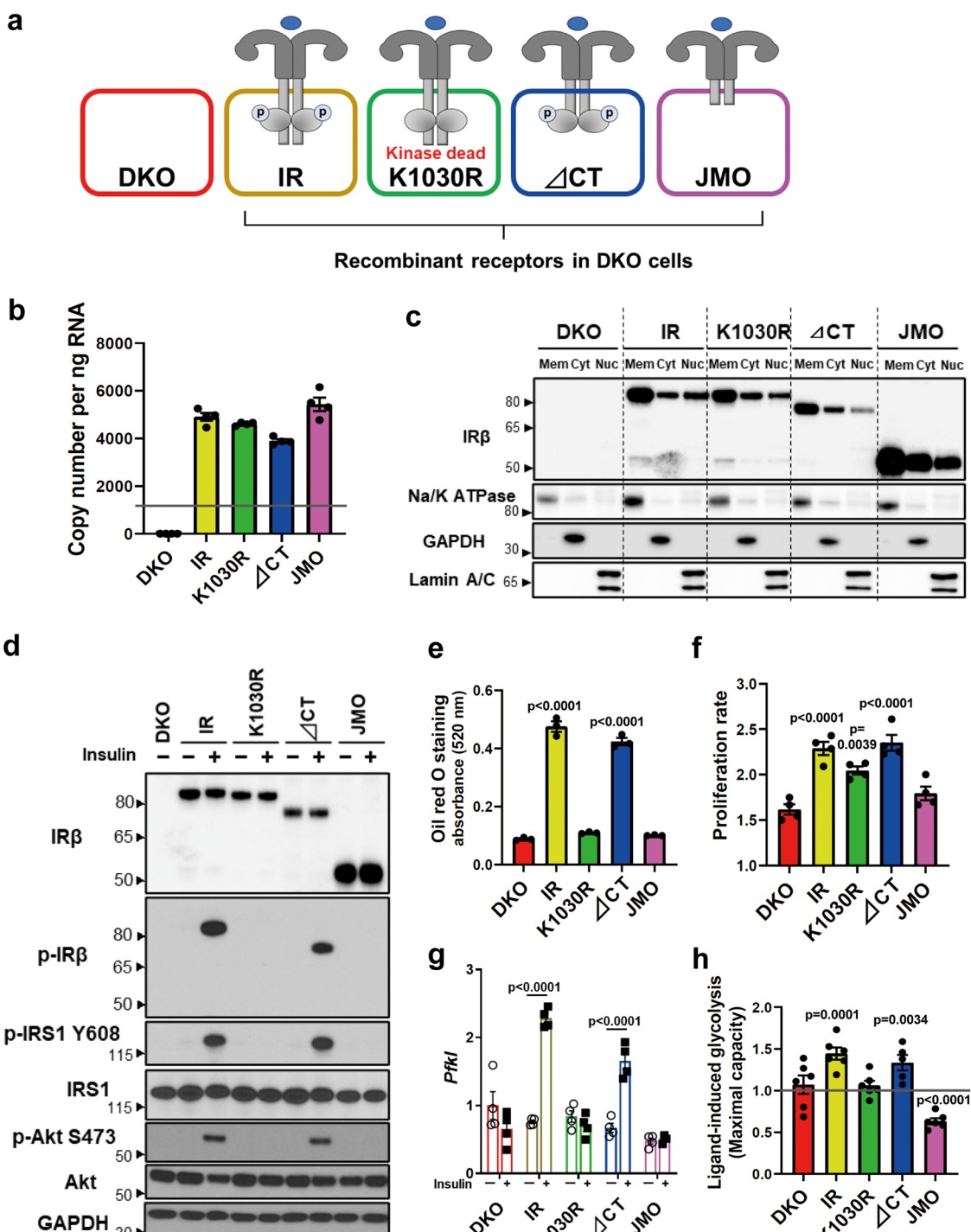

**Fig. 1 | Kinase- and domain-dependent roles of IR in regulating metabolism and growth. a** Schematic representation of DKO preadipocytes reconstituted with IR, K1030R mutated IR (K1030R), mutated IR lacking 79 amino acids from the C-terminus (ΔCT), and mutated IR lacking most of the intracellular domain, but with the 36 amino acid juxtamembrane domain (JMO). **b** Relative mRNA levels of recombinant receptors of IR, K1030R, ΔCT, and JMO as determined by qPCR using cDNA standards for quantitation. The dotted line is the level of IR mRNA in WT cells. Data are means ± SEM copy number per ng total RNA (*n* = 4). **c** Immunoblotting of IR-beta subunit (IRβ) in lysates of the membrane (Mem), cytosolic (Cyt), and nucleus fraction from DKO, IR, K1030R, ΔCT, and JMO cells. **d** Immunoblotting of phosphorylated and total receptor protein levels, IRS-1$^{Y608}$ phosphorylation, and Akt$^{S473}$ phosphorylation in lysates from DKO, IR, K1030R, ΔCT, and JMO cells stimulated with 100 nM insulin for 15 min. **e** Triglyceride accumulation in DKO, IR,

K1030R, ΔCT, and JMO cells by Oil red O staining day 7 after induction of differentiation (*n* = 3). Data are means ± SEM. *P*-values vs. DKO, one-way ANOVA. **f** Proliferation rates of DKO, IR, K1030R, ΔCT, and JMO cells (*n* = 4) per day are shown as means ± SEM. *P*-values vs. DKO, one-way ANOVA. **g** mRNA levels of *Pfkl* in DKO, IR, K1030R, ΔCT, and JMO cells. Cells were FBS starved for 5 h with DMEM containing 0.1% BSA, then stimulated with or without 100 nM insulin for 6 h. mRNA levels of genes were analyzed by qPCR. Data are means ± SEM (*n* = 4). Gene expression levels of DKO cells at the basal were set at 1. *P*-values are basal vs insulin, two-way ANOVA. TBP expression was used to normalize gene expression. **h** Glycolytic rate (maximal glycolytic capacity) induced by insulin (100 nM) stimulation for 6 h. Fold change of the maximal glycolytic capacity as measured by ECAR on insulin stimulation was calculated for each cell line. Data are means ± SEM (*n* = 5–6). *P*-values are basal vs insulin stimulation.

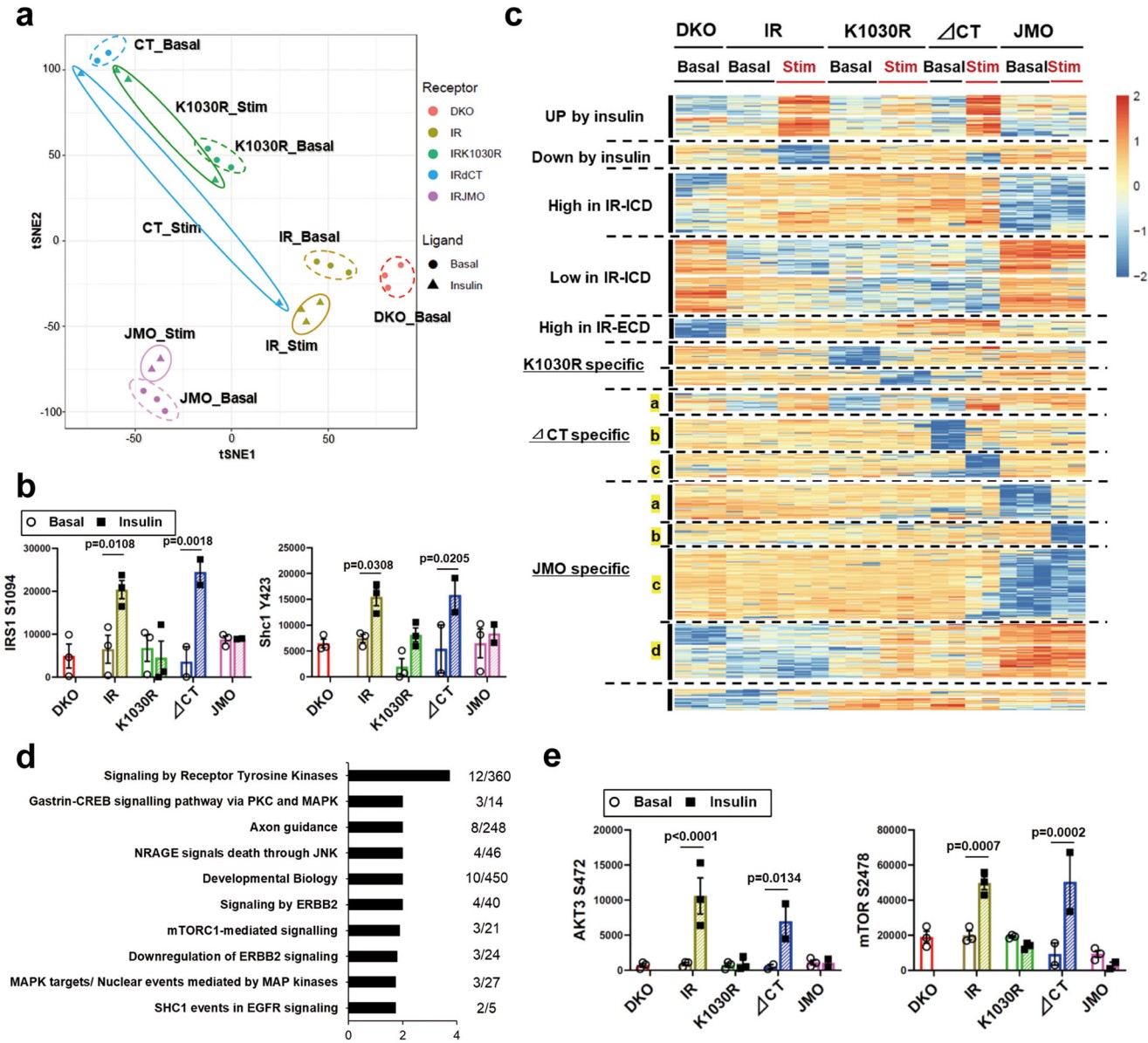

**Fig. 2 | Differential phosphoproteomic signatures of DKO, IR, K1030R, ΔCT, and JMO cells. a** T-distributed stochastic neighbor embedding (t-SNE) analysis of the phosphosites identified by LC–MS/MS from DKO, IR, K1030R, ΔCT, and JMO cells in the basal and insulin-stimulated states. Cells were serum starved for 6 h with DMEM containing 0.1% BSA before 100 nM insulin stimulation for 15 min. **b** Quantification of exemplary phosphorylation events in IRS1 and Shc1. Data are means ± SEM of phosphosites intensity values (×10⁴) (*n* = 2–3). *P*-values are basal vs. insulin, two-way ANOVA followed by Šídák's multiple comparisons test. **c** Heatmap showing the hierarchical clustering of the phosphopeptides in DKO, IR, K1030R,

ΔCT, and JMO cells in the basal and insulin-stimulated states. Values are *Z*-scores of log2 transformed intensity values. **d** REACTOME pathway enrichment analysis of phosphosites in the Up by Insulin cluster. The functional enrichment analysis was tested by the STRING database, where FDRs were calculated using the Benjamini–Hochberg procedure. Plots are −log10 transforms of enrichment FDR value. **e** Quantitation of exemplary phosphosites in the Up by Insulin cluster. Data are means ± SEM of phosphosites intensity values (×10⁴) (*n* = 2–3). *P*-values are basal vs insulin, two-way ANOVA.

state, and these were given imputed values as indicated in Methods. T-distributed stochastic neighbor embedding (t-SNE) analysis indicated the clear separation of the phosphoproteome in these five cell lines both in the basal state and after insulin stimulation (Fig. 2a). As expected, phosphorylation of proteins related to IR kinase-mediated signaling, such as IRS-1^S1094 and Shc1^Y423, were upregulated by insulin stimulation in IR and ΔCT cells but not in K1030R or JMO cells (Fig. 2b).

Heatmap analysis with hierarchical clustering demonstrated the specific phosphorylation pattern observed for each IR type and changes in the basal and the insulin-stimulated states (Fig. 2c). A total of 1963 phosphosites were significantly regulated among five cell lines under either the basal or the stimulated state (FDR < 0.05). It is worth

noting that if we had limited our analysis to phosphopeptides that have at least 80% observed values in all samples, this would reduce the number of both the number of phosphosites and the proportion of missing values by about half (7944 phosphopeptides with 9% missing values), but the resulting heatmap would be similar in pattern to that Fig. 2c (see Supplementary Fig. 2b for comparison). Of the differentially regulated phosphosites in Fig. 2c, 150 phosphosites on 114 proteins were upregulated by insulin and 73 phosphosites on 58 proteins were downregulated by insulin in cells expressing the normal IR or ΔCT IR, i.e., were receptor kinase activity-dependent phosphorylations. As expected, pathway analysis of these phosphoproteins in the "Up by Insulin" cluster revealed many pathways known to be linked to insulin

action, including signaling by receptor tyrosine kinases, MAP kinase, JNK, and mTORC1-related pathways (Fig. 2d). For example, insulin significantly upregulated phosphorylation of Akt3$^{S472}$ and mTOR$^{S2478}$ in IR and ΔCT cells, but not in other cells (Fig. 2e). The phosphorylation of Akt3$^{S472}$ as assessed by phosphoproteomics gave similar results to the western blot data of Akt phosphorylation in Fig. 1d and Supplemental Fig. 2c, in which the phospho-Akt antibody used can recognize Akt1 (S473), Akt2 (S474), and Akt3 (S472). In addition, 72 phosphosites on 62 proteins were upregulated by insulin in cells expressing the ΔCT IR > wild-type IR (subgroup a in the "ΔCT specific" cluster). Gene ontology analysis showed that this cluster was significantly enriched for proteins with molecular function-related MAP kinase activity (Supplementary Fig. 2d, e); this is consistent with previous reports that phosphorylation of the C-terminus of the IR may inhibit mitogenic signaling[11,12]. Cells expressing the ΔCT receptor also showed 117 ligand-upregulated and 90 ligand-downregulated sites (subgroups b and c in the "ΔCT specific" cluster), which were not regulated in cells expressing any of the other receptors. Pathway analysis of this "ΔCT specific" cluster included proteins involved in the cell cycle and RNA transport. For example, BRCA1$^{S717}$, a protein involved in mRNA nuclear export, was upregulated by insulin, and UPF1$^{S1102}$, a protein involved in mRNA surveillance and initiation of nonsense-mediated mRNA decay, was down-regulated by insulin in a manner dependent on the receptor C-terminal domain (Supplementary Fig. 2f, g).

## Phosphorylation signatures independent of ligand and kinase activity

In addition to the insulin-regulated sites, the unbiased cluster analysis revealed many phosphosites that were independent of the kinase activity of IR and not regulated by insulin. Thus, 221 phosphosites on 147 proteins were increased in cells expressing the intact IR or K1030R and ΔCT mutants compared to DKO cells, but not in cells expressing the JMO-IR. These constituted a "High-Phos in IR-intracellular domain (IR-ICD)" cluster. Similarly, we identified 264 phosphosites on 203 proteins whose phosphorylation was low in cells expressing IR or IR-variants with most ICD (IR, K1030R, ΔCT receptor) compared to DKO cells, i.e., a "Low-Phos in IR-ICD" cluster.

Pathway analysis of proteins in the "High-Phos in IR-ICD" cluster showed enrichment in proteins involved in cell junction organization and cell–ECM interactions (Supplementary Fig. 3a). Examples of phosphosites in these pathways, including phosphorylations on filamin-C, NCK2, and plectin (PLEC), etc. are quantified in Fig. 3a and Supplementary Fig. 3b. On the other hand, pathway analysis of the "Low-Phos in IR-ICD" cluster included cell cycle and mitosis related sites, such as MCM6$^{S13}$ and ENSA$^{S2}$ (Fig. 3b, c). This cluster also included proteins in the ATM serine/threonine kinase and its downstream targets (Fig. 3d, e and Supplementary Fig. 3c). ATM is a Ser/Thr protein kinase which is normally activated by DNA double-strand breaks and phosphorylates several key proteins involved in cell cycle arrest, DNA repair, apoptosis and the cellular senescence phenotype[20] (Fig. 3d). Phosphorylation of ATM$^{S1897}$, Chk2$^{S264}$, 53BP1$^{S418}$, and RNF168$^{S197}$, all important components of the ATM signaling pathway, were down-regulated by 30–60% in cells expressing IR with a full or near-full intracellular domain, regardless of whether or not it contained an active tyrosine kinase and independent of the absence or presence of insulin (Fig. 3e and Supplementary Fig. 3c). In addition to these alterations, several phosphosites on proteins associated with inflammatory signaling and cellular senescence, such as IFNAR2$^{S403}$, NFKB1$^{S447}$, and JAK2$^{S523}$ were also downregulated in cells with an IR-ICD (Supplementary Fig. 3d). In addition, the "Low-Phos in IR-ICD" cluster included phosphosites on Small Ubiquitin-like Modifier (SUMO)/E3 ligases/SUMOylation target proteins, such as PML$^{S528}$, SP100$^{S314}$, and SMC1A$^{S360}$ (Fig. 3f, g). SUMOylation is a post-translational modification involved in various cellular processes, such as transcriptional regulation, apoptosis, protein stability, senescence, and response to

stress[21,22]. The SUMOylation process and SUMOylation of promyelocytic leukemia protein (PML), which is a major constituent of the PML nuclear body (NB), including SP100, etc., are known to induce cellular senescence[21]. In addition, PML-NBs are emerging as coactivators of cellular genes that exert antiviral activities, such as cytokines and interferon (IFN)-stimulated genes (ISGs)[22]. This cluster was also enriched in phosphosites on proteins associated with membrane trafficking, such as SHKBP1$^{S10}$, KIF26B$^{S1689}$, and PAFAH1B2$^{S2}$ (Fig. 3h). Interestingly, SH3KBP1 binding protein 1 (SHKBP1) has been shown to modulate signaling from the EGF receptor tyrosine kinase[23]. While no common kinase could be identified that regulated these phosphorylations in a kinome analysis[24,25], the "Low-Phos in IR-ICD" cluster showed enrichment of phosphosites that are targets of protein phosphatase 1A (PPM1A), protein tyrosine phosphatase type IVA 3 (PTP4A3) and several dual-specificity phosphatases (DUSPs) (Supplementary Fig. 3e). PPM1A, PTP4A3 and DUSP1 have been shown to be involved in the induction of cell-cycle arrest[26,27].

The phosphoproteome also included other phosphorylation clusters, including regulated phosphorylations specific to the K1030R receptor and phosphorylations specific to the JMO receptor (Fig. 2c). Cells expressing the K1030R (tyrosine kinase-dead) receptor showed 72 ligand-upregulated and 54 ligand-downregulated sites, which were not regulated in cells expressing any of the other mutated receptors and wild-type IR. Several proteins in this cluster were associated with the cell cycle, such as 53BP1 (Supplementary Fig. 3f, g). A large cluster containing 704 phosphosites on 490 proteins was increased or decreased phosphorylation uniquely by IR-JMO ("JMO specific" in Fig. 2c) and is discussed later in this paper.

## Kinase- and domain-dependent roles of IR in gene expressions

To understand how the signaling events related to different domains of IR can result in altered cell function, we assessed gene expression in the DKO, IR, K1030R, ΔCT, and JMO cells using RNAseq after 6 h serum starvation with or without stimulation by 100 nM insulin for 15 min (Supplementary Data 2). Principal component analysis (PCA) and heatmap analysis with hierarchical clustering indicated that cells expressing the intact IR and each of the mutated IRs clearly separated from DKO cells and that each had a unique gene expression pattern (Fig. 4a, b). Using pair-wise moderated t-tests with an FDR of <0.05 and a cutoff of at least a 1.5-fold difference in expression, 1130 genes were significantly differentially expressed between groups in either the basal or ligand-stimulated state. Due to the short time of stimulation (15 min), only 32 genes were upregulated by insulin in cells expressing wild-type IR. All of these were also upregulated in ΔCT cells, but not the cells expressing other receptors (Fig. 4c). This cluster contained many well-known early response genes/transcription factors, including *Egr1–3*, *Fos*, *Jun*, *Jun-b*, *Atf3*, and *Hes1*[18]. This cluster also contained some genes related to important cellular processes, including DNA-damage response (*Ier3* and *Ppp1r15a*), regulation of RNA stability (*Zfp36*), and cytoskeletal effects on synaptic plasticity (*Arc*). Insulin induction of *Fos* and *Hes1* was confirmed by qPCR (Fig. 4d).

Surprisingly, the majority of genes (745 genes) differentially up or downregulated in IR, K1030R, and ΔCT-expressing cells as compared to DKO cells were dependent on the intracellular domain, i.e., were lost in JMO cells, but independent of ligand stimulation or having an active kinase site (Fig. 4b). Of these, 285 were increased in cells expressing the intact IR, or K1030R or ΔCT mutants both in the basal and stimulated state, but were low in DKO cells and cells expressing the JMO-IR. These constituted a "High Expression in IR-ICD" cluster. Similarly, 460 genes were low in cells expressing IR or IR-variants with most or all of the ICD (IR, K1030R, ΔCT receptor), but were high in DKO and JMO-IR cells; these constituted a "Low Expression in IR-ICD" cluster. Pathway analysis of the "High Expression in IR-ICD" genes showed significant enrichment for genes related to cholesterol biosynthesis, ECM organization, and collagen synthesis (Fig. 5a). These included 2- to 3-fold

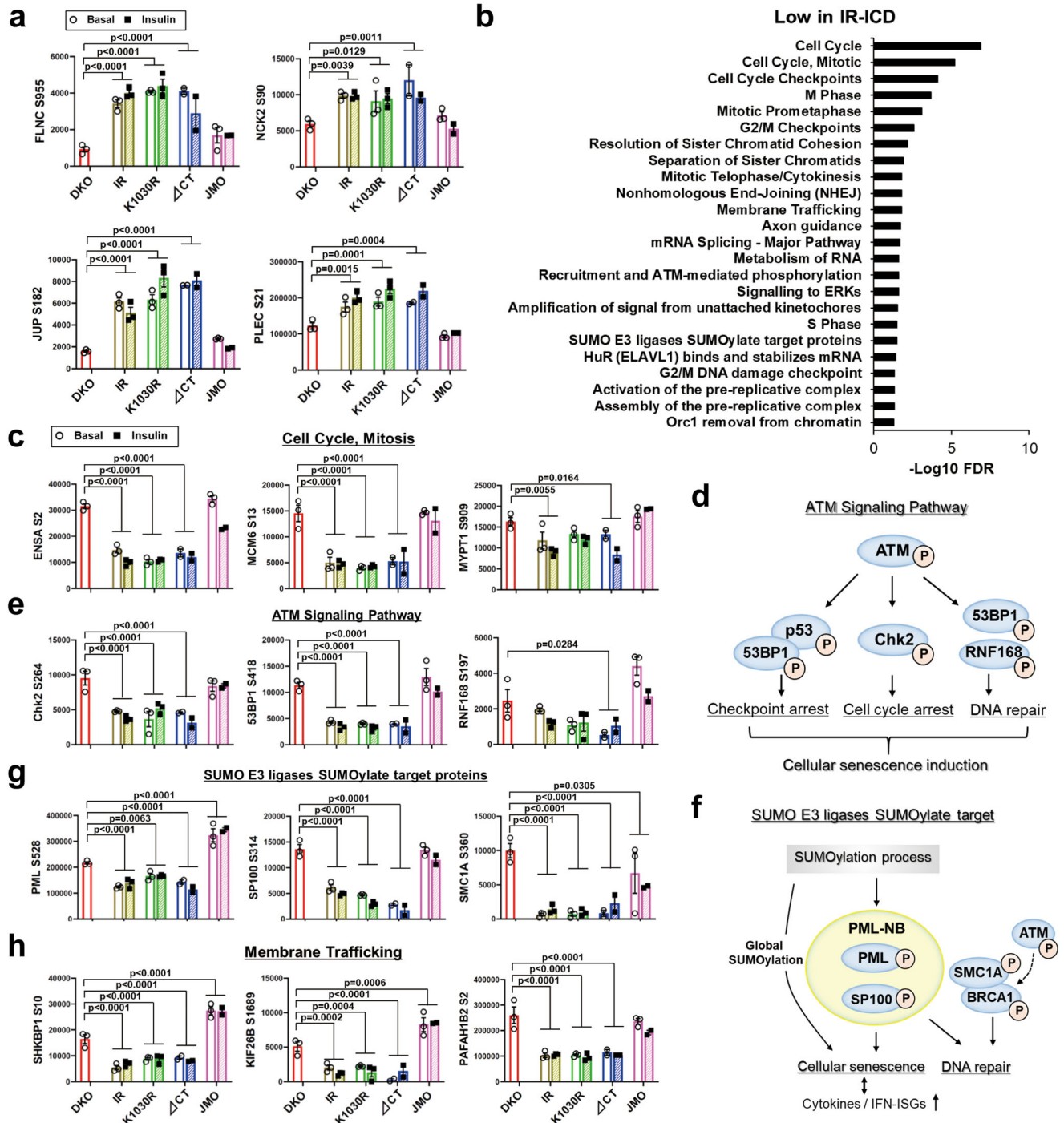

**Fig. 3 | Phosphorylation signatures are independent of kinase activity by DKO, IR, K1030R, ΔCT, and JMO cells. a** Quantitation of exemplary phosphosites in the enriched pathways (in Supplementary Fig. 3a) in the High-Phos in IR-ICD cluster. **b** REACTOME pathway enrichment analysis of phosphosites in the Low-Phos in IR-ICD cluster. The functional enrichment analysis was tested by the STRING database, where FDRs were calculated using the Benjamini–Hochberg procedure. **c** Quantification of exemplary phosphosites of Cell Cycle and Mitosis pathway in the Low-Phos in IR-ICD cluster. **d, e** Representation of the ATM signaling pathway

(**d**) and quantification of some important phosphosites in the pathway (**e**). **f, g** Representation of the "SUMO E3 ligases SUMOylate target proteins" pathway (**f**) and quantification of some important phosphosites in the pathway (**g**). **h** Quantification of exemplary phosphosites associated with membrane trafficking in the Low-Phos in IR-ICD cluster. Data in (**a, c, e, g, h**) are means ± SEM of phosphosite intensity values (×10⁴). *P*-values vs DKO (combined both basal and insulin for comparisons), one-way ANOVA followed by Dunnett's multiple comparisons test (*n* = 3–6).

increases in multiple collagen genes (*Col1a1, Col3a1,* and *Col5a1*) and up to 2-fold increases in key genes regulating cholesterol synthesis including *Hmgcs1* and *Sqle* (Fig. 5b). On the other hand, the cluster "Low Expression in IR-ICD" was significantly enriched for genes related to the immune system, especially related to IFN and JAK/STAT1/2 stimulated genes (IFN-SGs) (Fig. 5c). Examples of these with

quantification are shown in Fig. 5d and Supplementary Fig. 4a; most with changes in gene expression ranging from 2- to almost 10-fold. While several of these IFN-SGs, such as *Oas2, Ifit1,* and *MX1*, are known to have anti-viral activities[28,29], in the absence of viral infection, these IFN/immune-related pathways are closely associated with cellular senescence and the senescence-associated secretory phenotype

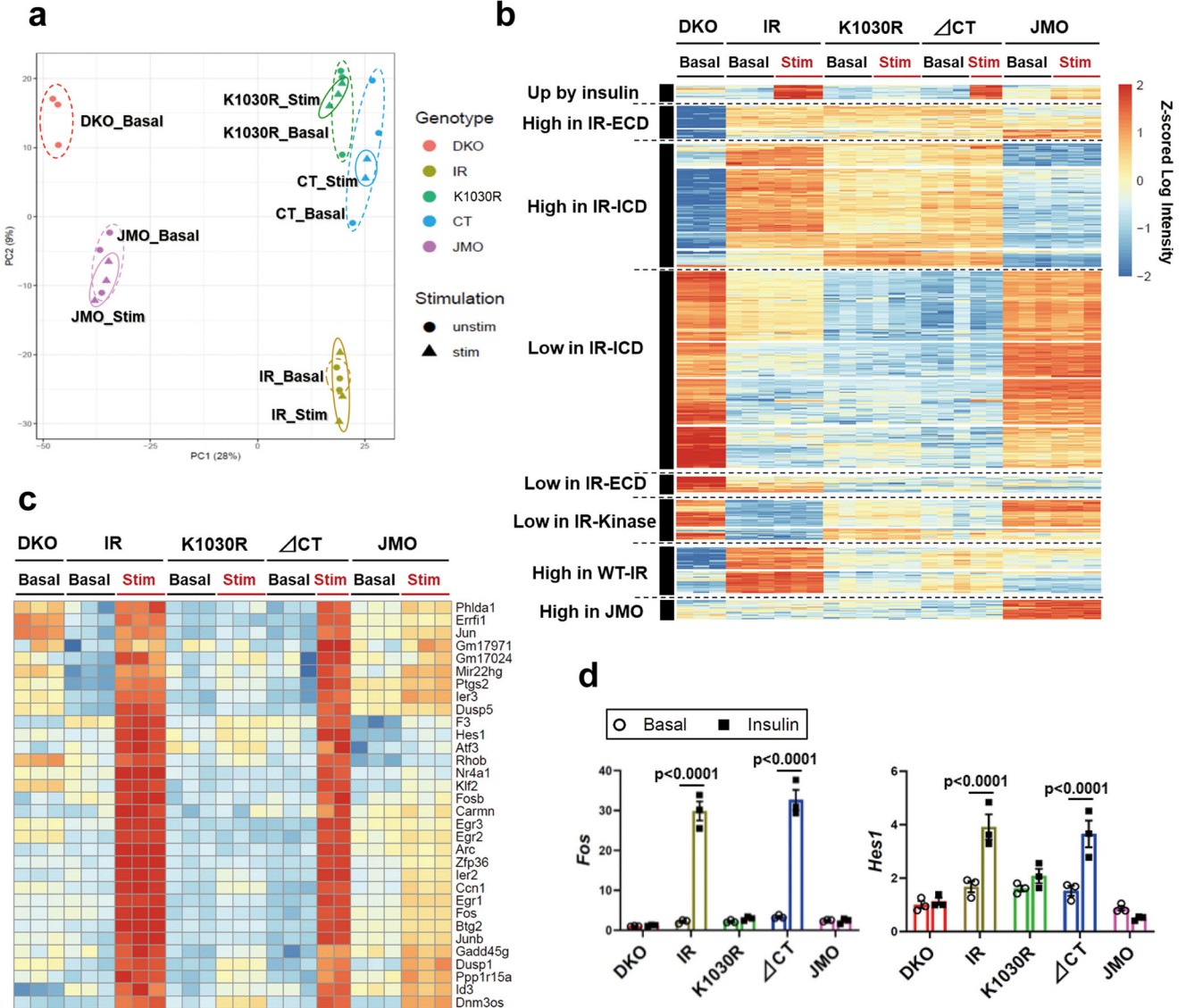

**Fig. 4 | Kinase- and domain-dependent roles of IR in gene expressions.**
**a** Principal component analysis (PCA) of transcriptomes from DKO, IR, K1030R, ΔCT, and JMO cells. Cells were serum starved for 6 h with DMEM containing 0.1% BSA before 100 nM insulin stimulation for 15 min. **b** Heatmap showing the hierarchical clustering of regulated genes in DKO, IR, K1030R, ΔCT, and JMO cells in the basal and insulin-stimulated states. Values are Z-scores of log2 transformed intensity values. **c, d** Heatmap of all upregulated genes by insulin stimulation in both IR and ΔCT cells (**c**) and confirmation of exemplary genes by qPCR (**d**). P-values are basal vs insulin, two-way ANOVA followed by Šídák's multiple comparisons test (n = 3).

(SASP)[30,31]. Further analysis of these data revealed that the levels of *IL-6, Ccl2, Cxcl10*, and *Mmp3*, also well-known SASP-related genes[31,32], were down-regulated by 30–80% in cells expressing IR with a near full-length ICD, again independent of the presence of ligand or an active kinase domain (Fig. 5e). Mechanistically, this decrease in SASP-related and ISGs may be due to decreases in IFNs themselves, since mRNA levels of IFN-α and IFN-β, which are upstream of JAK/STAT1/2-ISGs signaling, were downregulated in IR-ICD expressing cells by 70-90% as assessed by qPCR (Fig. 5f).

We also analyzed changes in mRNA expression of selected genes in these cell lines following long-term insulin stimulation (100 nM insulin for 6 h) by qPCR (Supplementary Fig. 4b–f). Well-known insulin-regulated genes, such as *Srebp1c* and *Pfkl*, were significantly upregulated by insulin only in IR and ΔCT cells (Supplementary Fig. 4b and Fig. 1g). mRNAs for genes in the "High Expression in IR-ICD" cluster, i.e., collagens and cholesterol synthesis related genes, such as *Fdps* and *Sqle*, were slightly upregulated in IR and ΔCT cells by insulin (Supplementary Fig. 4c, d), while mRNA levels of IFNs and IFN-ISGs,

representing genes in the "Low Expression in IR-ICD" cluster, showed little change following insulin stimulation (Supplementary Fig. 4e, f).

The heatmap in Fig. 4b also included other smaller clusters, such as genes with "High Expression in IR-ECD", "Low Expression in IR-ECD", "High Expression in WT-IR only," and "High Expression in JMO only" clusters (Supplementary Fig. 4g–k). The "High Expression in WT-IR" cluster contained genes involved in ECM organization, such as integrin subunit alpha-8 (*Itga8*) and calpain 6 (*Capn6*) (Supplementary Fig. 4j, k).

## Roles of different intracellular domains of IR on protein expression

A limited correlation between mRNA and protein levels is a well-known biological phenomenon[33], but to what extent this is modulated by metabolic cues is less well understood. Therefore, to determine to what extent the changes in gene expression resulted in alterations of protein levels in cells expressing different functional states of IR, we performed proteomic analysis under the same conditions as in the

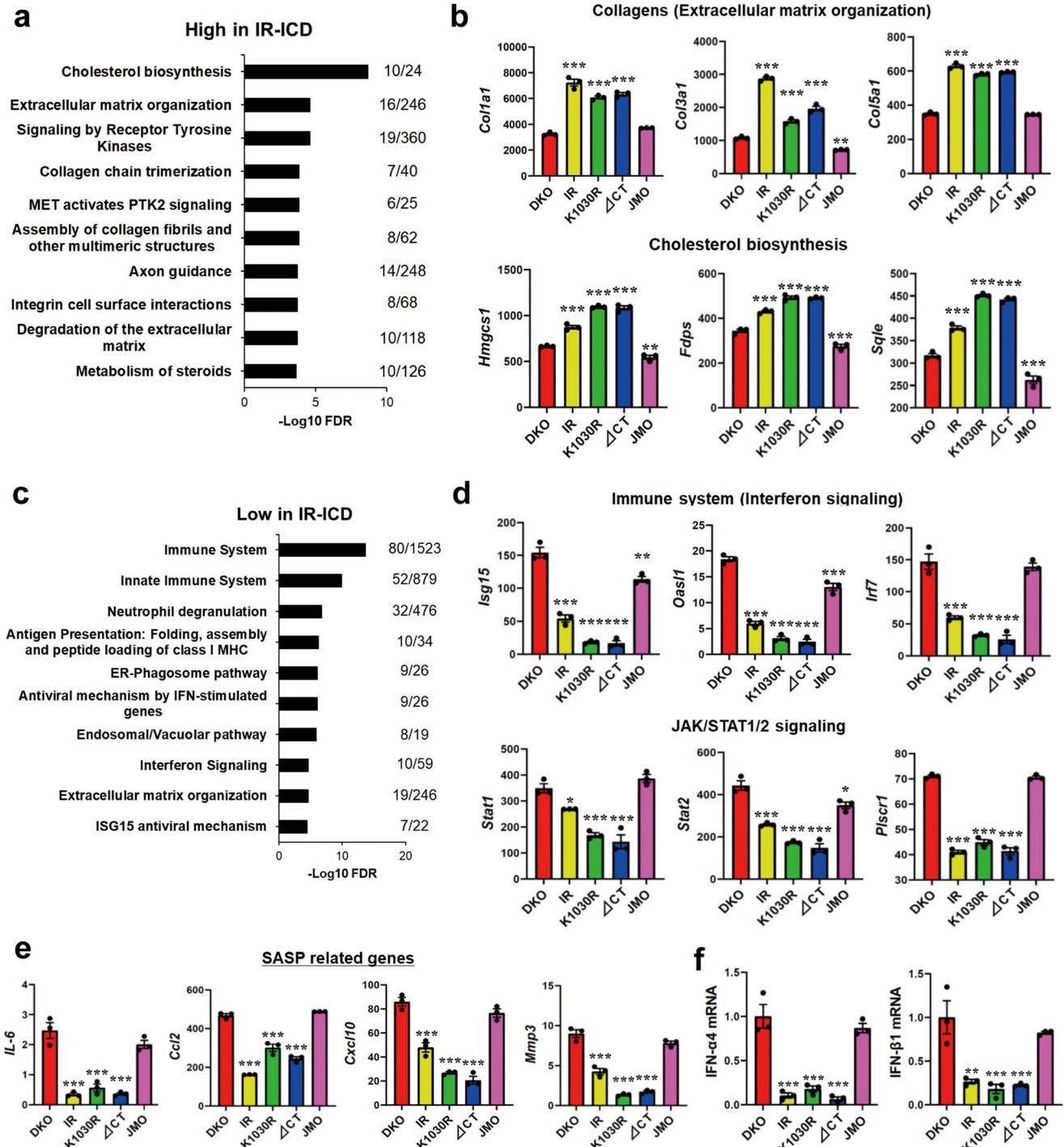

**Fig. 5 | Analysis of IR kinase-independently regulated genes. a** Pathway analysis showing the top 10 upregulated REACTOME pathways by IR-ICD. The functional enrichment analysis was tested by the STRING database, where FDRs were calculated using the Benjamini–Hochberg procedure. Plots are –log10 transforms of enrichment FDR value. **b** Quantification of exemplary genes in important pathways from the high expression in IR-ICD cluster at the basal. Data are means ± SEM of genes intensity values ($n = 3$). **$P < 0.01$, ***$P < 0.001$ vs. DKO, one-way ANOVA. **c** Pathway analysis showing the top ten downregulated REACTOME pathways by IR-ICD. The functional enrichment analysis was tested by the STRING database, where

FDRs were calculated using the Benjamini–Hochberg procedure. **d** Quantitation of exemplary genes in important pathways from the low expression in IR-ICD cluster. Data are means ± SEM of genes intensity values ($n = 3$). *$P < 0.05$, **$P < 0.01$, ***$P < 0.001$ vs. DKO, one-way ANOVA. **e** Quantitation of some genes related to senescence-associated secretory phenotype (SASP). Data are means ± SEM of gene intensity values ($n = 3$). ***$P < 0.001$ vs. DKO, one-way ANOVA. **f** Gene expression levels of IFN-α4 and IFN-β1 by qPCR. Data are means ± SEM. The level in DKO was set at 1. **$P < 0.01$, ***$P < 0.001$ vs. DKO, one-way ANOVA ($n = 3$).

RNA sequencing (Supplementary Data 3). As expected, IR and IGF1R proteins were not detected in DKO cells (Supplementary table 1). As with gene expression, PCA of proteomic data revealed that cells expressing IR, K1030R, and ΔCT had overlapping clusters, whereas

DKO and JMO cells formed unique groups (Fig. 6a), indicating a high degree of similarity in the proteome of the cells with the structurally intact intracellular domain of the IR, independent of whether the IR has kinase activity, but distinct proteomic changes in cells lacking both IR

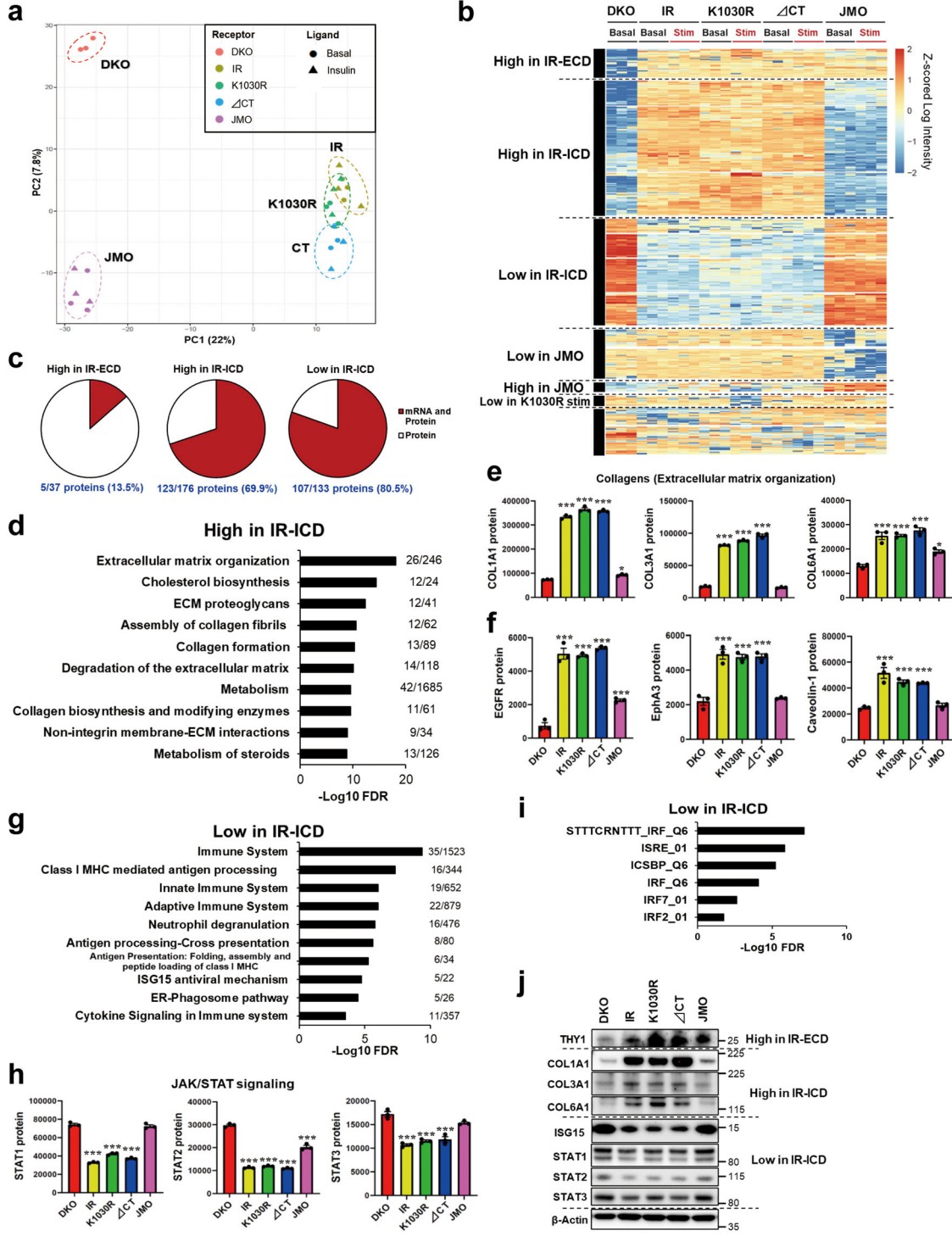

**Fig. 6 | Proteomic analysis of DKO, IR, K1030R, ΔCT, and JMO cells. Analysis of IR-ICD-dependent up and downregulated proteins. a** PCA of the proteome from DKO, IR, K1030R, ΔCT, and JMO cells. Cells were serum starved for 6 h with DMEM containing 0.1% BSA before 100 nM insulin stimulation for 15 min. **b** Heatmap showing the hierarchical clustering of regulated proteins in DKO, IR, K1030R, ΔCT, and JMO cells in the basal and insulin-stimulated states. Values are Z-scores of log2 transformed intensity values. **c** Percentage of proteins with the same changes in mRNA expressions in the high protein expression in IR-ECD, the high protein expression in IR-ICD, and the low protein expression in IR-ICD clusters. **d** Pathway analysis showing the top 10 REACTOME pathways in the high protein expression in the IR-ICD cluster. Plots are −log10 transforms of enrichment FDR value. **e** Quantification of exemplary proteins of collagens from the high protein expression in IR-ICD cluster at the basal. Data are mean ± SEM of protein intensity values (n = 3). *P < 0.05, ***P < 0.001 vs. DKO, one-way ANOVA. **f** Quantification of

some proteins-related signaling receptors in the high protein expression in IR-ICD cluster at the basal. Data are mean ± SEM of protein intensity values (n = 3). ***P < 0.001 vs. DKO, one-way ANOVA. **g** Pathway analysis showing the top 10 REACTOME pathways in the low protein expression in the IR-ICD cluster. **h** Quantitation of exemplary proteins in the JAK/STAT signaling pathway from the low protein expression in IR-ICD cluster at the basal. Data are means ± SEM of proteins intensity values (n = 3). ***P < 0.001 vs. DKO, one-way ANOVA. **i** Enrichment analysis of gene motifs in the low protein expression in IR-ICD cluster tested using the Fisher exact test. Plots are −log10 transforms of enrichment FDR value. **j** Immunoblotting of THY1, COL1A1, COL3A1, COL6A1, ISG15, STAT1-3, and β-Actin in lysates from DKO, IR, K1030R, ΔCT, and JMO cells. The functional enrichment analyses in (**d**, **g**) were tested by the STRING database, where FDRs were calculated using the Benjamini–Hochberg procedure.

and IGF1R and with IR lacking most of the intracellular domain. Using pair-wise moderated *t*-tests to detect proteins that were differentially expressed by at least 1.5-fold, a total of 489 proteins were significantly different between groups in either the basal or stimulated states at an FDR of <0.05. These are shown in a hierarchical clustering heatmap in Fig. 6b. As expected, there was no ligand response due to the short period of ligand stimulation. Consistent with the PCA, the majority of proteins in the heatmap were regulated by the ICD of IR, but independent of having an active kinase domain. When combined with the transcriptome analysis, 70–80% of proteins in the "High Protein Expression in IR-ICD" and the "Low Protein Expression in IR-ICD" clusters were regulated in parallel with mRNA expression (Fig. 6c).

In agreement with the correlation between mRNA and protein, pathway analysis showed that the "High Protein Expression in IR-ICD" cluster was significantly enriched for proteins related to ECM organization, collagens, and cholesterol biosynthesis (Fig. 6d). Examples of proteins in these pathways with quantification are shown in Fig. 6e and Supplementary Fig. 5a and include collagens COL1A1, COL3A1, and COL6A1, several of which were increased up to fivefold, in good agreement with the mRNA expression data. This also agreed well with changes in the phosphorylation of known regulators of collagen expression. Thus, the phosphorylation of JNK and total amounts of NFKB1 protein, both of which are negative regulators of collagen gene transcription[34], were higher in DKO and JMO cells than in IR, K1030R, and ΔCT cells (Supplementary Fig. 5b), as were other genes downstream of JNK, such as *Jun* and *Mmp8* (Supplementary Fig. 5c, d). Interestingly, this cluster which was increased in expression in cells with a near full-length IR-ICD, but independent of kinase activity, also included PDGF receptor binding (Supplementary Fig. 5e), several other signaling receptors [EGFR, EphA3, EphB3, and DDR2 (discoidin domain receptor tyrosine kinase 2)], β1-integrin, and caveolin-1, a membrane scaffolding protein that can alter receptor tyrosine kinase signal transduction and trafficking[35] (Fig. 6f and Supplementary Fig. 5f).

Likewise, consistent with the transcriptomic data, the "Low Protein Expression in IR-ICD" cluster was significantly enriched for proteins related to the immune system, especially cytokine signaling (including IFN related signaling and STAT3) and JAK/STAT1/2-ISGs signaling (Fig. 6g, h and Supplementary Fig. 5g). This suppression is likely due to transcriptional regulation since motif analysis of genes/proteins in this cluster showed enrichment for IFN-regulatory factor 7 and 2 (IRF7 and IRF2) transcription factor motifs (Fig. 6i). Further supporting this notion, the mRNA level of IRF7 was lower in cells expressing an IR with a full-length or near full-length ICD (Fig. 5d). Some important proteins regulated by ECD and ICD of IR were confirmed by immunoblotting (Fig. 6j and Supplementary Fig. 5h). There was also a unique cluster of proteins whose expression was high in cells expressing wild-type IR and all three mutated receptors i.e., the high expression is either related to the presence of either the IR extracellular domain (ECD) or possibly the transmembrane or juxtamembrane region of the receptor. Only 5 out of the 37 (14%) of the proteins in this cluster were similarly regulated as the mRNA level (Fig. 6c and Supplementary Fig. 5i, j), indicating that these changes represented primarily post-transcriptional regulation.

Taking the proteomic data along with the phosphoproteomic data shown in Fig. 2, we were able to evaluate the stoichiometry of phosphorylation for the 9750 sites in proteins that were identified in both analyses. Although the total number of phosphosites that could be analyzed was reduced by about 35%, the protein-normalized phosphoproteomic data, as represented in the t-SNE plot and heatmap analysis, were very similar to those of the total phosphoproteome (Supplementary Fig. 6), indicating that most of the phosphorylation changes were the result of changes in the stoichiometry of phosphorylation, not changes in protein abundance. Pathway analysis revealed that the presence of receptors with most of the intracellular domain (IR-ICD), whether kinase active or inactive, was associated with changes in phosphorylation of a network of proteins involved in the regulation of ECM organization, cell cycle, mitosis, and ATM signaling (Supplementary Fig. 6c–g), similar to the total phosphoproteome analysis (Figs. 2 and 3). In the ATM signaling pathway (Fig. 3d), while Chk2 and RNF were not identified in the proteomic data, ATM$^{S1897}$ and 53BP1$^{S418}$ showed similar changes in the normalized phosphoproteome (Supplementary Fig. 6g).

## JMO-dependent signature in signaling and gene/protein expression

In addition to the pathways controlled by the presence of the larger IR-ICD, reconstitution of DKO cells with the JMO-IR, which contains an IR with a 36 amino acid intracellular domain attached to the transmembrane and ECDs, could create some unique signals. Thus, cells expressing the JMO-IR exhibited a unique group of up and down-regulated phosphorylation events involving 704 sites on 490 proteins, which differed from those observed in DKO cells or cells expressing all other receptors (Fig. 2c). Pathway analysis showed that this "JMO specific" phosphorylation cluster was significantly enriched for proteins related to cell cycle, mRNA splicing, SUMOylation, membrane trafficking, histone demethylation and cell–cell communication (Fig. 7a–c). Kinase enrichment analysis of the phosphosites revealed enrichment of protein kinase A catalytic subunit sites.

In parallel, some genes and proteins were uniquely regulated by JMO-IR. At the mRNA level, JMO-IR cells showed upregulation of 46 genes, including several cell cycle-related genes (*Cdkn1a* and *Trp53inp1*) (Fig. 7d). The proteomic heatmap (Fig. 6b) also included 11 proteins in a "High Protein Expression in JMO" and 61 proteins in a "Low Protein Expression in JMO" clusters. The latter was significantly enriched for proteins related to negative regulation of the immune system process and regulation of cell differentiation (Fig. 7e, f). These findings demonstrate how different portions of the intracellular domain, and possibly even the ECD, of the IR, can contribute to distinct signaling events and altered gene and protein expressions in the regulation of cellular function.

## LYK-I signaling contributes to cellular sensitivity to apoptosis

Since proteins related to IFN signaling through JAK/STAT and NFKB pathways are closely associated with the regulation of apoptotic sensitivity and cellular senescence, and since several of these were altered in gene/protein expression and phosphorylation in cells in an LYK-I-dependent manner, we measured DNA fragmentation and caspase activation in WT, DKO, IR, K1030R, ΔCT, and JMO cells under several conditions known to induce apoptosis (Fig. 8a, b and Supplementary Fig. 7). While DNA fragmentation was increased in all cell lines by serum starvation, $H_2O_2$ treatment, and etoposide treatment, DKO cells showed marked reduction in DNA fragmentation compared to WT cells under all conditions. Interestingly, the re-introduction of the wildtype IR, K1030R, or ΔCT mutant receptors, but not the JMO-IR, largely rescued the apoptotic response to all three stimuli as measured by DNA fragmentation (Fig. 8a). Appearance of cleaved caspase 3, another marker of apoptosis, in these samples showed similar results (Fig. 8b and Supplementary Fig. 7a). Thus, apoptotic sensitivity of cells was rescued by the presence of an IR with a full-length or near full-length ICD, but this effect was independent of IR tyrosine kinase activity. Cell viability following $H_2O_2$ or etoposide treatment showed similar results, i.e., cells expressing wildtype IR or the K1030R mutant IR showed significantly decreased cell survival rate following $H_2O_2$/etoposide treatments as compared to DKO and JMO cells, with the ΔCT-IR giving an intermediate response (Fig. 8c), indicating that the intracellular domain, but not the kinase activity of the IR, provide signals required for normal regulation of stress-induced apoptosis. Apoptotic sensitivity of cells was also rescued by re-expression of mouse IR (mIR) in DKO cells (Supplementary Fig. 8a–c). Re-expression of mIR in DKO cells also

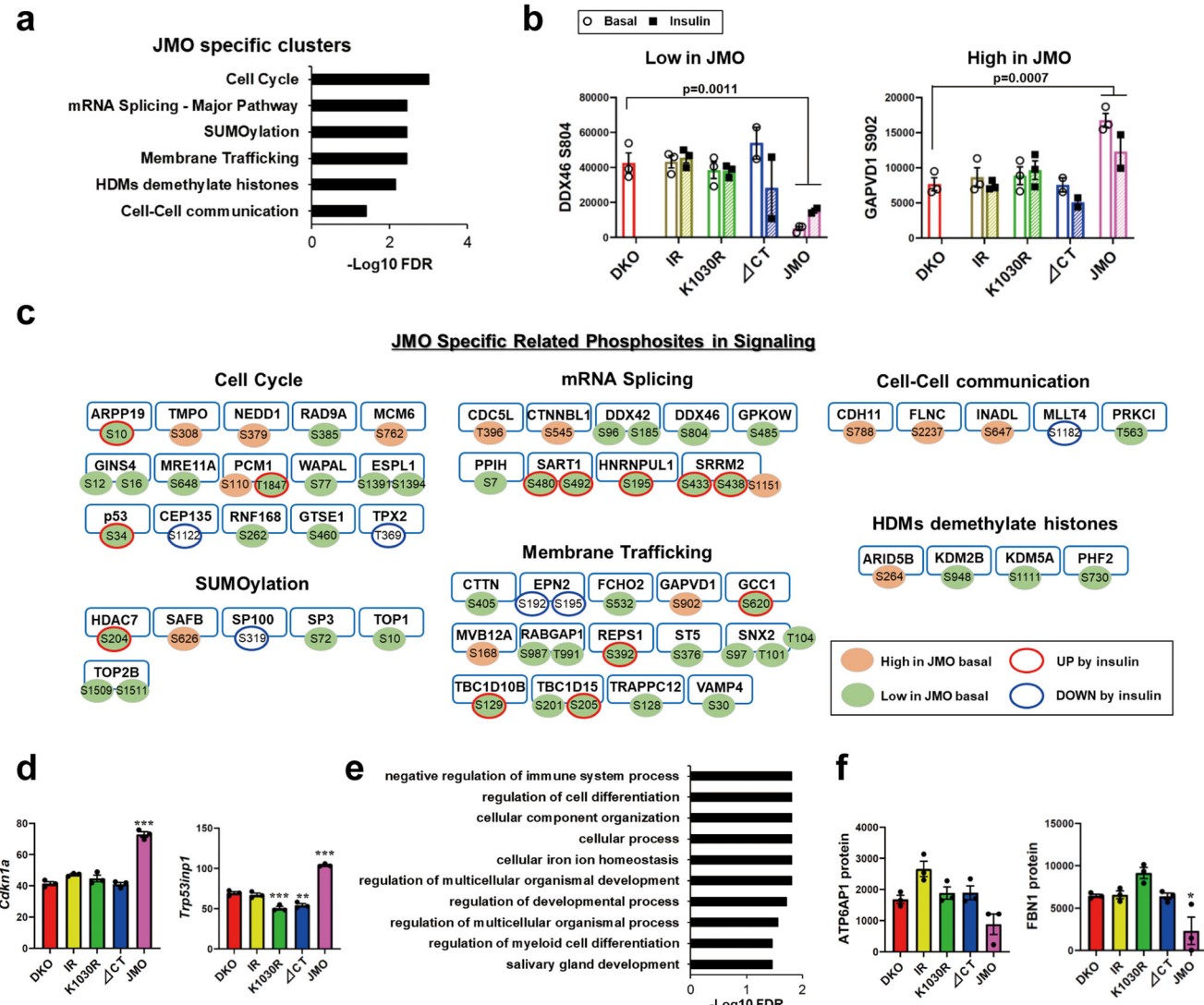

**Fig. 7 | JMO dependent signature in signaling and gene and protein expressions. a** REACTOME pathway enrichment analysis of phosphosites regulated by JMO-IR. The functional enrichment analysis was tested by the STRING database, where FDRs were calculated using the Benjamini−Hochberg procedure. Plots are −log10 transforms of enrichment FDR value. **b** Quantification of exemplary phosphosites in the enriched pathways (in (**a**)) in the "JMO specific" cluster. Data are means ± SEM of phosphosites intensity values (×10⁴). *P*-values vs. DKO (combined both basal and insulin for comparisons), one-way ANOVA followed by Dunnett's multiple comparisons test (*n* = 3–6). **c** Diagram of intracellular signaling regulated by JMO-IR as identified by phosphoproteomics. Each of the phosphosites was color-coded based on the effects of basal or insulin stimulation on phosphorylation. **d** Quantification of exemplary genes in the high expression in JMO cluster in the RNA-sequencing heatmap at the basal. Data are means ± SEM of genes intensity values (*n* = 3). \*\**P* < 0.01, \*\*\**P* < 0.001 vs. DKO, one-way ANOVA. **e, f** The top 10 enriched GO term (Biological process) (**e**) and quantification of exemplary proteins (**f**) in the low protein expression in JMO cluster in the proteome heatmap at the basal. The functional enrichment analysis in (**e**) was tested by the STRING database, where FDRs were calculated using the Benjamini−Hochberg procedure. Data are means ± SEM of proteins intensity values (*n* = 3). \**P* < 0.05 vs. DKO, one-way ANOVA.

showed increased mRNA expressions of collagens and decreased mRNA expressions of *IFN-α4* and *Isg15* (Supplementary Fig. 8d). This resistance to apoptosis was mediated by activation of the JAK/STAT/IFN-SGs and NFKB1 pathways in DKO and JMO cells, pathways known to be associated with resistance to apoptosis[29,36]. Thus, apoptosis resistance, as indicated by decreased levels of cleaved caspase 3 in DKO cells, was rescued by treatment with TPCA-1, an inhibitor of both NF-κB pathway and STAT3 and RNAi-based knockdown of STAT3 or both STAT3 and NFKB1 (Fig. 8d and Supplementary Fig. 7b). On the other hand, treatment with baricitinib, an inhibitor of JAK/STAT signaling (Fig. 8d) or knockdown of STAT1 or NFKB1 alone did not alter cleaved caspase 3 levels (Supplementary Fig. 7c). Taken together with the above data, these data indicate that the LYK-I effects on gene and protein expression play important roles in the regulation of cellular senescence and apoptosis through a complex series of

phosphorylation changes which are, at least in part, related to ATM, STAT3 and NFKB1 signaling (Fig. 8e).

## Discussion

The IR is activated by its cognate ligand and, through autophosphorylation on critical tyrosine residues within its kinase domain, initiates its canonical signaling pathway[4,37]. This is mediated through phosphorylation of intracellular substrates (IRS proteins, Shc, and others), which activate a network of serine/threonine phosphorylation resulting in a broad spectrum of metabolic and growth effects, including regulating glucose transport, lipid synthesis, gluconeogenesis, and glycogen synthesis, and control of cell cycle and growth[4]. In states of insulin resistance, many of these actions are reduced, contributing to the pathogenesis of various diseases, including type 2 diabetes mellitus and metabolic syndrome[38]. However, recent studies

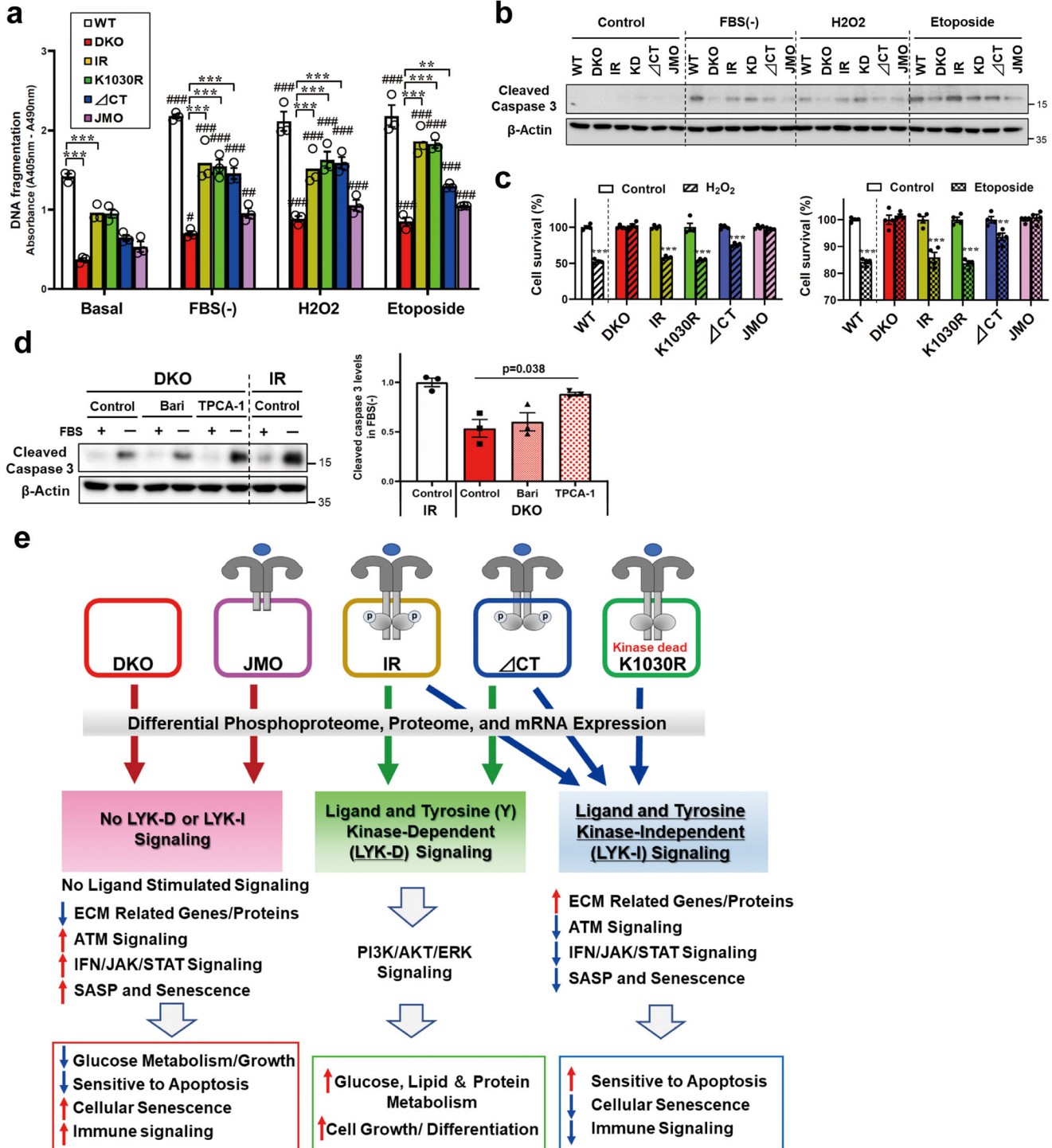

**Fig. 8 | LYK-I signaling contributes to cellular sensitivity to apoptosis.**
**a**, **b** Changes in DNA fragmentation (**a**) and immunoblotting of cleaved caspase 3 (**b**) in WT, DKO, IR, K1030R, ΔCT, and JMO cells following serum starvation for 6 h, 500 μM $H_2O_2$ for 24 h or 25 μM etoposide for 24 h ($n = 3$). Data are means ± SEM. **$P < 0.01$, ***$P < 0.001$ vs. DKO, #$P < 0.05$, ##$P < 0.01$, ###$P < 0.001$ vs. basal, two-way ANOVA. **c** Cell viability in WT, DKO, IR, K1030R, ΔCT, and JMO cells treated with 2 mM $H_2O_2$ or 50 μM etoposide for 24 h using an Alamar Blue assay ($n = 4$). Data are means ± SEM. **$P = 0.002$, ***$P < 0.0001$, control vs. treatment. **d** Immunoblotting of cleaved caspase 3 levels in lysate from DKO and IR cells with or without 6 h FBS starvation. Before FBS starvation, cells were treated with baricitinib (JAK-1/2 inhibitor) or TPCA-1 (IKK inhibitor) for 48 h ($n = 3$). Data are means ± SEM. The level in

the IR control was set at 1. Comparisons between groups were performed using one-way ANOVA. **e** Schematic model of LYK-D and LYK-I signaling. In addition to the classical ligand and tyrosine kinase-dependent (LYK-D) signaling pathways, the insulin receptor can mediate many receptor-dependent, but ligand- and kinase-independent (LYK-I) signaling events. These LYK-I actions of the IR are associated with changes in phosphorylation of a network of proteins involved in the regulation of extracellular matrix organization, cell cycle, ATM signaling, and cellular senescence and result in upregulation of expression of multiple extracellular matrix-related genes and proteins; down-regulation of immune/interferon-related genes and proteins; and increased sensitivity to apoptosis.

have suggested that the effects of the IR may not be limited to classical kinase-dependent actions and that the unoccupied IR may also exert effects on sensitivity to apoptosis and expression of some imprinted genes and miRNAs[10,13,14]. In the present study, we sought to determine the mechanisms underlying such ligand-independent IR signaling and its biological functions using a combination of functional studies and "omics" approaches in cells expressing no endogenous IR and IGF1R (DKO) and cells expressing the normal human IR, a kinase-dead IR (K1030R), an IR lacking the C-terminal 79 amino acids (ΔCT) and an IR with only 36 juxtamembrane intracellular amino acids, thus lacking both the kinase domain and C-terminus (JMO).

We find that in addition to the classical LYK-D pathways that can be stimulated or suppressed by insulin, IRs with a complete or near complete intracellular domain can also regulate multiple biologically important pathways in a kinase and ligand-independent manner. Thus, compared to DKO cells, cells expressing the intact IR or a kinase-dead IR (K1030R) exhibit a broad network of changes in cellular protein phosphorylation. Most of these receptor kinase-independent changes are also observed following expression of the ΔCT-IR, but not the JMO-IR, indicating that in addition to containing the site of tyrosine kinase activity, the region of the IR between amino acids 989 and 1276 contains a second, tyrosine kinase-independent signaling function. These phosphorylation events are also ligand-independent, thus we have termed these actions ligand and tyrosine kinase-independent (LYK-I) actions and the associated region of the receptor the LYK-I domain.

LYK-I activity regulates the phosphorylation of 485 sites on 341 proteins, located primarily in the cytoplasmic compartment of the cell, although some are also found in the nucleus. The phosphosites downregulated as part of these LYK-I actions are on proteins regulating cell cycle progression, mitosis, ATM-mediated signaling, SUMOylation, and cellular senescence. It is worth noting that while the phospho-proteomic analysis includes multiple phosphopeptides in the same phosphosite, and that there was some variability as to the magnitude of the differences observed depending on the multiplicity state analyzed, most of the changes were quite robust. The net result of these changes is higher rates of cell proliferation in cells containing IR with the LYK-I domain compared to DKO and JMO cells. In addition, many genes and proteins are up or downregulated by LYK-I actions of the IR. The upregulated genes include genes involved in ECM organization, multiple collagen species, and genes of cholesterol biosynthesis. This difference in collagens might be due to higher levels of JNK and NFKB1 signaling in DKO and JMO cells since JNK and NFKB1 are negative regulators of collagen transcriptions[39]. Collagen fibers play a role in signal transmission-related to cell proliferation and differentiation[40,41]. Cellular senescence has been linked to decreased collagen expressions[42].

In addition to these up-regulated genes/proteins, 460 genes and 133 proteins are downregulated by the presence of an IR with an LYK-I domain. These genes/proteins include many proteins involved in the immune system, cytokine/IFN signaling, and JAK/STAT signaling. The JAK/STAT and NFKB1 signaling pathways are most often activated by cytokines and IFNs and play a role in the development and immune regulation[28-30] but have also been shown to have a role in mediating cellular senescence[30]. Stimulation of IFN receptors activates JAK/STAT signaling to result in the formation of STAT1/2 heterodimers, which associate with IRF9 to form IFN-stimulated gene factor 3 (ISGF3), which can bind to specific IFN-stimulated response elements (ISREs) and regulate transcription of IFN-stimulated genes (ISGs)[28,43]. Compared to DKO and JMO cells, cells expressing an IR with an LYK-I domain have lower mRNA levels of IFN-α and IFN-β and downregulation of IFN-stimulated genes, including *Isg15*, *Oasl1*, and *Irf7*, and SASP related genes[44,45]. Higher expression of IFNs, ISGs, and SASP-related genes, which are observed in DKO and JMO cells, are often regarded as hallmarks of senescent cells[43,44], suggesting that these LYK-I actions might reduce cellular senescence.

Many studies have shown that senescent cells are remarkably resistant to apoptosis[46], and some of the down-regulated proteins in cells expressing the IR-LYK-I domain, such as those noted above, as well as STAT3 and NFKB1, are associated with the inhibition of apoptosis. We have shown that DKO cells lacking both IR and IGF1R exhibit resistance to serum-depletion-induced and extrinsic factor-induced apoptosis, and this can be largely restored by re-expression of IR or the kinase-inactive K1030R-IR [present study and[13]]. This resistance to apoptosis in DKO cells can be rescued not only by wildtype IR and K1030R, but also by the ΔCT-IR, but not by the JMO-IR, pointing to a domain in the intracellular portion of the β-subunit of the IR involved in the LYK-I response. This resistance to apoptosis and DNA damage in DKO and JMO cells is also reversed by siRNA knockdown of STAT3 and NFKB1 or inhibitors of their signaling pathways. These pathways are closely related to the ability to respond to viral infections, and many downstream genes of JAK/STAT1/2-ISGs signaling, including *Oas2, Ifit1*, and *MX1*, which have anti-viral activities[28,29]. These data suggest that DKO and JMO cells might be more resistant to viral infection when compared to cells with an IR-LYK-I domain, a subject worthy of further investigation. Resistance to apoptosis is one of the hallmarks of cellular adaptations in cancer and autoimmune disease[47,48], as well as a feature of normal cellular senescence[46].

Apoptosis is an evolutionarily conserved cell death pathway that is important for normal development, maintenance of tissue homeostasis, and cancer prevention[49]. In addition, heightened apoptotic sensitivity is a hallmark of developing tissues. Since apoptosis enables the removal of damaged or superfluous cells, active cell proliferation during development is tightly coupled to the expression of the machinery necessary for apoptosis induction[49]. Our data show that LYK-I signaling by the IR can reverse apoptosis resistance, including resistance to both intrinsic and extrinsic stimuli. Thus, the LYK-I affect to increase apoptotic sensitivity might be important in the proper development and maintenance of tissue homeostasis. It may also be important in conditions such as starvation and exposure to oxidative stress. How the downregulation of IR number in different pathophysiological conditions[50] may affect this LYK-I response should be a subject for further study.

While the precise molecular link between the IR and the LYK-I effects needs further exploration, LYK-I signaling could result from direct effects of the receptor in different compartments as it migrates through the cell and/or interaction of the IR with other proteins in the cell in a kinase-independent fashion. We and others have shown that the IR can interact with proteins as varied as glypican-4[51], membrane metalloendopeptidase[52], the α-arrestin Arrdc3[53], and various β-arrestins[54,55]. Another candidate for LYK-I actions is NCK2 since cells expressing receptors with a near full-length ICD (LYK-I domain) show an increase in phosphorylation of NCK2 on Ser[90], which is independent of IR kinase activity. NCK2 is a member of the NCK family of adapter proteins and has no known catalytic functions, but does contain one SH2 and three SH3 domains, all of which can interact non-covalently with a variety of other proteins. The SH2 domain of NCK2 can interact with phosphotyrosine residues in the PDGF, VEGF, and EGF receptors, while the SH3 domains of NCK2 have been shown to interact IRS-1[56], as well as with signaling effectors containing proline-rich regions that mediate their activation by upstream kinases[57,58]. Interestingly, several RTKs, including EGFR, PDGFRα, EphA3, EphB3, and DDR2, as well as cell membrane proteins known to interact with RTKs, such as β1-integrin and caveolin-1, are significantly upregulated in cells expressing receptors associated with LYK-I actions. Further studies will be needed to determine if any of the receptors, receptor-associated proteins, or related docking proteins which are upregulated in cells exhibiting LYK-I actions could create this second signaling pathway.

Another potential mechanism of LYK-I effects could be direct transcriptional control by the IR or fragments of the IR in the nucleus. IRs are localized in caveolae and undergo clathrin-mediated

internalization[59]. Some fraction of IR also migrates to the nucleus[60]. Hancock et al. have shown that IR associates with promoters in DNA and have suggested that this interaction, acting through some yet-to-be-defined transcription factors, may regulate gene expression[61]. Exactly what domain or structures in the IR are required for these effects are not known. By cell fraction, we find evidence of a small fraction of wildtype IR and all of the IR mutants used in this study, including the JMO-IR, in the nucleus, even in the absence of insulin stimulation. These receptors in the nucleus in the basal and stimulated states acting with transcription factors and other proteins could potentially contribute to the regulation of transcription[60,62]. To what extent these internalized receptors may be involved in the regulation of gene expression or the LYK-I effects observed with the different receptor mutants remains to be determined.

In summary, we find that, in addition to the classical LYK-D signaling pathways, the IR can mediate many receptor-dependent, but LYK-I, signaling events. These previously unrecognized events are based on the presence of a domain of the β-subunit of the IR between residues 989 and 1276. This domain can upregulate expression of a large number of ECM-related genes and proteins, including multiple collagens and fibrillins, and can negatively regulate many immune-related genes and proteins, including those involved in cytokine, IFN, and JAK/STAT signaling, independent of its kinase activity. Together, these LYK-I actions of the IR regulate multiple important cellular functions, including cell cycle progression, apoptosis sensitivity, and inhibition of a cellular senescence phenotype with decreased secretion of SASP proteins. Thus, IRs, through the presence of the LYK-I domain, contribute a range of tyrosine kinase-independent signaling events, in addition to the classical ligand and tyrosine kinase-dependent events. While at present, we have studied these effects only in a single cellular model (mouse preadipocytes), further studies will help elucidate if these LYK-I actions are similar in other cell types or whether different cell types have a unique program of LYK-I effects. These LYK-I effects may play unique roles in biology and pathophysiology and provide approaches to the treatment of diseases in which these pathways are altered.

## Methods
All experimental protocols described in this study were approved by Harvard University's Committee on Microbiological Safety (Tissue-Specific Inactivation of the IR Gene in Mice: 21-054).

## Materials
Antibodies against phospho-IRβ/IGF1Rβ (#3024), IRβ (#3025), IGF1Rβ (#3027), β-Actin (#4970), p-AKT$^{S473}$ (#4060), AKT pan (#4685), p-JNK$^{T183/Y185}$ (#4668), JNK (#9252), NFKB1 (#12540), THY1 (#13801), COL1A1 (#91144), Lamin A/C (#2032), Cleaved Caspase-3 (#9661), STAT1 (#9172), STAT2 (#72604), STAT3 (#4904) and GAPDH (#5174) were purchased from Cell Signaling. Anti-COL6A1 antibody (sc-377143) and anti-ISG15 (sc-166755) antibodies were purchased from Santa Cruz. Anti-Phospho-IRS-1 (Y608) antibody (#09-432) and anti-IRS-1 antibody (#06-248) were purchased from Millipore. Anti-Na/K ATPase antibody (#ab7671) and anti-collagen III (#ab7778) antibody were purchased from Abcam. All antibodies were used at a dilution of 1:1000 except for β-Actin and GAPDH, which were used at dilutions of 1:5000. The Human insulin was purchased from Sigma. The human IR (B isoform) retroviral plasmid was generated previously in the lab[18]. IR$^{K1030R}$ (aa numbers excluding signal peptide) were generated from the human IR (B isoform) cDNA using a site-directed mutagenesis kit from Agilent. Primers for IR$^{K1030R}$ generation are 5′- GGCAGAGACCCGCGTGGCGGTG AGGACGGTCAACGAGTCAGCC-3′ and 5′- GGCTGACTCGTTGACCGTCC TCACCGCCACGCGGGTCTCTGCC-3′. For co-immunoprecipitation assays, human IR, IR$^{K1030R}$, ΔCT$^{1276}$-IR, and ΔCT$^{988}$-IR cDNA were cloned into the 3_Flag-CMV-14 mammalian expression vector (Sigma). siRNAs for mouse STAT1 (L-058881-00-0005), mouse STAT3 (L-040794-01-

0005), mouse NFKB1 (L-047764-00-0005) and non-targeting control (D-001810-10-05) were purchased from Horizon.

## Brown preadipocytes isolation, retroviral infection, and culture
IR/IGF1R double knockout brown preadipocytes (DKO cells) and DKO cells re-expressing mouse IR were used and described in our previous studies[9,10]. Briefly, preadipocytes were isolated from newborn homozygous IR-lox/IGF1R-lox mice by collagenase digestion of the brown fat pad and immortalized by infection with retrovirus encoding SV40 T-antigen followed by the selection with puromycin. The immortalized preadipocytes were infected with adenovirus expressing GFP alone to generate control cells (WT cells) or GFP-tagged Cre recombinase to generate double knockout (DKO) cells. GFP-positive cells were sorted and expanded in DMEM supplemented with 10% fetal bovine serum (FBS, Atlas Biologicals), 100 U/ml penicillin, and 100 μg/ml streptomycin (Gibco) at 37 °C in a 5% CO$_2$ incubator. IR/IGF1R double knockout preadipocytes were then stably transduced with pBabe retrovirus encoding human IR (B isoform), kinase-dead IR (K1030R mutation), ΔCT$^{1276}$-IR (ΔCT), ΔCT$^{988}$-IR (JMO) or empty vector. Plates (10 cm) of Phoenix cells were transiently transfected with 10 μg of pBabe-hygro retroviral expression vectors. Forty-eight hours after transfection, the virus-containing medium was collected and passed through a 0.45 μm pore size syringe filter. Filter-sterilized Polybrene (hexadimethrine bromide; 12 μg/ml) was added to the virus-loaded medium. This medium was then applied to proliferating DKO cells. 48 h after infection, cells were treated with trypsin and replated in a medium supplemented with hygromycin (Invitrogen) as a selection antibiotic. Cells were maintained in DMEM with 4.5 g/L glucose supplemented with 10% FBS, 1 mg/ml normocin (InvivoGen), 100 U/ml penicillin, and 100 μg/ml streptomycin (Gibco) and cultured in a humidified incubator at 37 °C and 5% CO$_2$. DKO brown preadipocytes re-expressing mouse IR (A isoform) were generated as described in our previous study[9]. Briefly, a mouse IR (MC224356) cDNA clone was purchased from Origene and subcloned into an empty pBabe-hygro vector. DKO cells were then stably transduced using this pBabe retrovirus to generate a mouse IR cell line[9]. All animal studies were approved by the Institutional Animal Care and Use Committee (IACUC) at the Joslin Diabetes Center and were in accordance with the National Institutes of Health guidelines.

## Quantification of mRNA Levels by qPCR and RNA-sequencing
Total RNA was isolated from cells using TRIzol reagent (Thermo Fisher Scientific). cDNA was synthesized and quantitative PCR amplification was conducted with the C1000 Thermal Cycler (BioRad, catalog CFX384) using iQ SybrGreen Supermix (Bio-Rad). The sequences of primers as follows: mouse IR, 5′-AAATGCAGGAACTCTCGGAAGCCT-3′ and 5′- ACCTTCGAGGATTTGGCAGACCTT-3′; mouse IGF1R, 5′-ATCGC-GATTTCTGCGCCAACA-3′ and 5′-TTCTTCTCTTCATCGCCCGCAGACT-3′; human IR, 5′-CGATATGGTGATGAGGAGCTGC-3′ and 5′-GTAGAAA-TAGGTGGGTTCCGTCCA-3′; mouse Fos, 5′-CGGGTTTCAACGCCGACTA-3′ and 5′-TTGGCACTAGAGACGGACAGA-3′; mouse Hes1, 5′-CCAGC-CAGTGTCAACACGA-3′ and 5′-AATGCCGGGAGCTATCTTTCT-3′; mouse IFN-α4, 5′-TGATGAGCTACTACTGGTCAGC-3′ and 5′-GATCTCTTAGCA-CAAGGATGGC-3′; mouse IFN-β1, 5′-CAGCTCCAAGAAAGGACGAAC-3′ and 5′-GGCAGTGTAACTCTTCTGCAT-3′; mouse Pfkl, 5′-GGAGGCGA-GAACATCAAGCC-3′ and 5′-CGGCCTTCCCTCGTAGTGA-3′; mouse TBP, 5′-ACCCTTCACCAATGACTCCTATG-3′ and 5′-TGACTGCAGCAAATCGC TTGG-3′.

For RNA sequencing, total RNA samples (3 μg) that passed the quality tests were submitted to the Biopolymers Facility at Harvard Medical School. The KAPA mRNA HyperPrep kit for Illumina sequencing was used. mRNA was pulled down using oligo-dT beads, and the resulting mRNA was converted into cDNA. The resulting cDNA then became a library through adapter ligation and post-PCR Cleanup. RNA-seq raw reads were 100-bp reverse-stranded single-end reads. The

reads were trimmed for adapters and filtered by sequencing Phred quality (≥Q15) using fastp[63]. The count table was generated by aligning reads to the mouse transcriptome (Ensembl version 98) using kallisto[64] and converting transcript counts to gene counts using tximport[65]. To filter out low-expressed genes, we kept genes that had more than 0.5 counts per million in at least 4 samples. Counts were normalized by the weighted trimmed mean of M-values[66]. To detect genes that were differentially expressed with at least 1.5-fold change between a pair of groups, we applied the TREAT method[67] using the R package limma[68] via our R package ezlimma[69], available on GitHub at https://github.com/jdreyf/ezlimma. P-values were corrected using the Benjamini–Hochberg false-discovery rate (FDR). The genes that had FDR < 0.001 in any pair-wise comparison were selected for detection of gene clusters in a hierarchical dendrogram using a variable cut height approach[70]. Heatmaps were created with the R package pheatmap. Pathway analysis of gene clusters was done using STRING[71] with default parameters.

## Cell lysate and immunoblotting

Cells were washed once with ice-cold phosphate-buffered saline (PBS) and collected in RIPA buffer (MilliporeSigma) supplemented with protease and phosphatase inhibitors (Biotool). Lysates were then centrifuged at 20,000×g for 15 min at 4 °C. Protein concentrations were measured by using the BCA protein assay (Thermo Fisher Scientific). Subcellular fractionation was done using the subcellular protein fractionation kit (Thermo Fisher Scientific) according to the protocol described by the manufacturer. Proteins were transferred to polyvinylidene difluoride (PVDF) membranes. Membranes were blocked in Starting Block T20 solution (Thermo Fisher Scientific) at room temperature for 1 h, washed three times with PBS with Tween-20, and incubated with an appropriate dilution of the primary antibody in Starting Block T20 solution overnight at 4 °C. The immunoblots were then washed three times with PBS with Tween-20 and incubated with an appropriate dilution (1:1000 or 1:5000) of horseradish peroxidase (HRP)-conjugated secondary antibody in Starting Block T20 solution for 1 h. The goat anti-rabbit HRP conjugated antibody (#1706515) was purchased from BioRad and the sheep anti-mouse HRP conjugated antibody (#NA931) was purchased from Millipore Sigma. Detection was achieved using the chemiluminescent detection reagents (Thermo Fisher Scientific).

## Adipocyte differentiation

Brown preadipocytes were differentiated as described previously[18]. Differentiated cells were maintained in DMEM with 4.5 g/L glucose, 1 mg/ml normocin, 100 units/ml penicillin, 100 g/ml streptomycin, and 10% FBS. On day 7 after differentiation, lipid accumulation was visualized by oil red O staining. Cells were washed once with PBS and fixed with 10% buffered formalin for 30 min. Cells were then stained with filtered oil red O solution (5 g/L in isopropyl) diluted 5/3 times in deionized-distilled water for 1 h at room temperature. After destaining each well with isopropyl, triglyceride accumulation in cells was evaluated by measurement of 520 nm absorbance.

## Ligand-induced glycolysis

The glycolysis rate of cultured cells was measured by Seahorse XFe96 Bioanalyzer (Agilent Technologies) according to the protocol of Agilent Seahorse XF Glycolysis Stress Test kit. Confluent cells were serum starved overnight, followed by 100 nM insulin for 5 h. Before analysis, cells were washed once with PBS, incubated in 175 μl XF base media (Agilent Technologies) supplemented with 2 mM glutamine with or without 100 nM insulin, and incubated at 37 °C without $CO_2$ for 1 h. ECAR values were measured in the basal, after injection of glucose, oligomycin, and a 2-deoxy-glucose. The maximal glycolytic capacity was calculated by ECAR value after oligomycin injection from the ECAR in the absence of glucose. The quantitative value was normalized by the respective protein content.

## Phosphoproteome and proteome analysis

For phosphoproteomics, samples were processed as described previously[10,19,72]. Totally, 750 μl acetonitrile (ACN) and 250 μl TK buffer (36% trifluoroacetic acid (TFA) and 3 mM KH2PO4) were added to the digested peptides, and the samples mixed in a ThermoMixer for 30 s at 1500 rpm. Debris was centrifuged at 18,000×g for 15 min, and the supernatant was transferred to 2 ml Deep Well Plates (Eppendorf). $TiO_2$ beads (prepared in 80% ACN, 6% TFA buffer) were added (1:10 ratio protein/beads) and then incubated in a ThermoMixer at 40 °C and 2000 rpm for 5 min. The $TiO_2$-bound phosphopeptides were pelleted by centrifugation, transferred to new tubes, and washed 4 times in wash buffer (60% ACN, 1% TFA) to remove non-specific or non-phosphorylated peptides. The beads were suspended in a transfer buffer (80% ACN, 0.5% acetic acid), transferred on top of single-layer C8 Stage Tips, and centrifuged until dry. The phosphopeptides were eluted with elution buffer (40% ACN, 20% NH4OH) and concentrated in a SpeedVac for 20 min at 45 °C. The phosphopeptides were then acidified by the addition of 100 μl of 1% TFA and loaded onto equilibrated SDBRPS (styrenedivinylbenzene-reversed phase sulfonated, 3 M Empore) Stage Tips. The SDBRPS StageTips were washed once in isopropanol/1% TFA and twice with 0.2% TFA. Finally, the desalted phosphopeptides were eluted with 60 μl of elution buffer (80% ACN, 1.25% NH4OH). The dried elutes were resuspended in an MS loading buffer (3% ACN, 0.3% TFA) and stored at −80 °C for LC-MS/MS measurement. LC-MS/MS was performed using Q Exactive HF-X Hybrid Quadrupole–Orbitrap Mass Spectrometer (Thermo Fischer Scientific) coupled online to a nanoflow EASY-nLC1200 HPLC (Thermo Fisher Scientific) as described previously[72]. These cell lysates in SDC buffer (4% SDC, 100 mM Tris pH 8.5) were also used for proteome analysis and immunoblotting.

Phosphopeptides enrichment, LC–MS/MS measurement, and phosphoproteomic data analysis were processed as described previously[10,72]. The raw files were processed using the Maxquant software environment (version 1.5.5.2) with the built-in Andromeda search engine for the identification and quantification of phosphopeptides. The data were searched using a target-decoy approach with a reverse database against the Uniprot proteome fasta file with a false discovery rate of 1% at the level of proteins, peptides, and modifications using minor changes to the default settings as follows: oxidized methionine, acetylation (protein N-term) and phospho was selected as variable modifications and carbamidomethyl as fixed modification. A maximum of 2 missed cleavages were allowed, a minimum peptide length of seven amino acids, and enzyme specificity was set to trypsin. The Maxquant output phosphoSTY table was processed using Perseus (version 1.5.2.11) software suite. Using all phosphopeptides, including multi-phosphorylated peptides, we filtered for class-I phosphosites (localization probability >0.75) that have at least 50% valid values in all samples measured. This yielded 14,971 phosphosites with 20.9% missing values. The remaining missing values were imputed using a quantile regression approach for the imputation of left-censored missing data (QRILC) from the R package imputeLCMD with default parameters. As a sensitivity analysis of our filtering paradigm, we examined the results using a paradigm in which we kept only phosphopeptides that had at least 80% observed values in all samples. This reduced both the number of analyzed phosphosites and the proportion of missing values by about half (7944 phosphopeptides with 9% missing values). The heatmap analysis corresponding to this filtering paradigm revealed 1704 significantly regulated phosphopeptides (Supplementary Fig. 2b), which appeared similar to the Fig. 2c heatmap and has a very significant overlap in the identified phosphosites ($p < 10^{-15}$ by one-sided Fisher exact test). The values were normalized

by quantile normalization[73] using the limma function normalize between arrays. To discover the differentially regulated phosphosites among all groups, we tested phosphosites with moderated $F$-tests in the R package limma[68]. $P$-values were corrected using the Benjamini−Hochberg FDR. Differences were considered significant when the FDR was <0.05. The phosphosites that had an FDR < 0.05 were selected for hierarchical cluster analysis using a variable cut height approach[70]. Heatmaps were created with the pheatmap R package. Protein sets were tested to determine if the phosphosites in each cluster are over-represented. The protein sets based on kinase substrates (PhosphositePlus[24] and RegPhos[25]) and phosphatases (the Dephosphorylation Database (DEPOD)[74], downloaded from Harmonizome[75]) were tested using the Fisher exact test via the Fisher_enrichment function from the R package ezlimma[69] using default parameters. Other protein sets were analyzed using STRING[71] for Mus musculus with default parameters. The functional enrichment analysis was tested by the STRING database[71], where FDRs were calculated using the Benjamini−Hochberg procedure. A detailed description of each analysis is in the "README" sheet in the supplemental phospho-proteome dataset.

In the proteome analysis, we kept proteins that had at least 50% valid values and imputed the remaining missing values using the QRILC method from the R package imputeLCMD. This yielded 4342 proteins with only 5.25% missing values. The values were then normalized by setting all sample medians to be equal. To discover proteins that are differentially regulated with at least 1.5-fold change between a pair of groups, we applied the TREAT method[67], as we did with the RNA-seq data, using the R package limma[68] via our R package ezlimma[69]. The proteins with FDR < 0.05 in any pair of groups were selected for detection of protein clusters in a hierarchical dendrogram using a variable cut height approach[70]. Protein sets were tested to determine if the proteins in each cluster are over-represented. Protein sets that belong to gene motifs (transcription factor or microRNA targets) from MSigDB[76] were tested using the Fisher exact test via the Fisher_enrichment function from the R package ezlimma[69] using default parameters. Other protein sets were analyzed using STRING[71] for Mus musculus with default parameters. A detailed description of each analysis is in the "README" sheet in the supplemental proteome dataset.

### Dealing with multiplicity in the phosphoproteome

Singly phosphorylated (multiplicity of 1): A phosphopeptide with a single phosphosite quantified example MAPK14_T180_M1 represents the singly phosphorylated peptide. The multiplicity of 2 (M2), MAPK14_T180_M2, and MAPK14_Y182_M2 share the same intensity as they represent the two phosphosites on the same peptide. The multiplicity of 3 (M3), indicates phosphosites located on phosphopeptides with more than two phosphorylations. The Maxquant tool reports the intensities of a phosphosite on peptides. For example, the intensity of MAPK14_T180_M1 and MAPK14_T180_M2 differs. For each phosphosite on multiply phosphorylated peptides, we receive a row with the same intensities as these phosphorylations are localized on the same peptide. While MAPK14_T180_M1 represents the singly phosphorylated peptide, MAPK14_T180_M2 and MAPK14_Y182_M2 share the same intensity as they represent the two phosphosites on the same peptide. Different phosphosites on the same peptide can have slightly different fold changes due to data filtering and imputation.

### Apoptosis and cell viability assays

DNA fragmentation was measured by using Cell Death Detection ELISAPLUS (MilliporeSigma) as described previously[13]. Cell viability under several conditions inducing apoptosis was assessed using alamarBlue Cell Viability Reagent (Invitrogen) according to the manufacturer's instructions.

### mRNA knockdown experiment

For siRNA treatment, cells were transfected with Lipofectamine RNAiMAX reagent (Thermo Fisher Scientific) and 25 nM siRNA according to the manufacturer's protocol. After 48 h incubation of the transfected cells, the culture medium was replaced with a normal medium containing 10% FBS or DMEM with 4.5 g/L glucose containing 0.1% BSA (serum starvation) for 6 h. Then, the cells were collected for the measurement of apoptotic sensitivity.

### Statistics and reproducibility

Data are presented as mean ± SEM. Comparisons between more than two groups were performed using one-way ANOVA followed by Dunnett's multiple comparisons test or Tukey's multiple comparisons test. Comparisons between two groups and two nominal variables were performed using two-way ANOVA followed by Sidak's multiple comparisons test. In all cases, the significant level was set at $p$-value < 0.05. Data are collected by using Microsoft Excel (Office 2016). All measures were replicated at least 2 times and, in most cases 3 or more (The two cell lines of the phosphoproteome and Supplementary Fig. 1c were done in duplicate).

### Reporting summary

Further information on research design is available in the Nature Portfolio Reporting Summary linked to this article.

## Data availability

Source data are provided in this paper. The RNA-sequencing data generated in this study have been deposited in the GEO database under accession code GSE206565. The mass spectrometry data generated in this study have been deposited in the PRIDE database under accession code PXD035587.

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

## Acknowledgements

This work was supported by NIH grants R01DK031036 (to C.R.K.) and the Joslin DRC grant (P30DK036836). H.N. was supported by a Sunstar Foundation postdoctoral fellowship and a JSPS Overseas Research Fellowship. W.C. was supported by NIH grants K01 DK120740 and P30 DK057521. A.K.J. and M.M. were supported by the Max Planck Society for the Advancement of Science and by the German Research Foundation (DFG/Gottfried Wilhelm Leibniz Prize).

## Author contributions

H.N. designed research, performed experiments, analyzed the data, and wrote the paper. A.K.J. performed proteome and phosphoproteomics analyses. W.C., T.M.B., and B.B.B. helped with experiments, and the generation of cell lines, and reviewed and edited the manuscript. H.P. and J.M.D. performed bioinformatics analysis. M.M. supervised the phosphoproteome analysis. C.R.K. designed the research, wrote the paper, and supervised the project.

## Competing interests

The authors declare no competing interests.
