## [Peer Review File · Nature Communications]

Reviewers' Comments:

Reviewer #1:

Remarks to the Author:

This manuscript uncovers a novel regulatory role of the insulin receptor intracellular domain. This tour de force omics study combines phosphoproteomics, RNAseq and proteomics to uncover novel functions of the insulin receptor beyond its normal ligand-kinase functions. While this work has the potential to be very exciting, as it suggests a new role of the insulin receptor in apoptosis and senescence, some concerns should be met before it can be considered for publication.

Notably, this entire work was conducted only utilizing a single cell line that was isolated from BAT and immortalized more than 12 years ago. While it was key to utilize the double knockout background for initial studies, some of the main effects of the paper need to be validated in other cell lines to demonstrate that these findings are generalizable beyond this one unique cell line. Thus, it will be helpful to repeat some of the main findings using re-expression of mouse insulin receptor as these effects may result from expressing a mutant human protein into a mouse cell line.

A large portion of the discussion restates the results section, instead the authors may wish to discuss why they believe the insulin receptor would evolve these alternative functions and importantly promote apoptosis?

All graphs should show individual data points. Specially figure 8 and figure 1.

Reviewer #2:

Remarks to the Author:

The authors have previously reported that the Insulin Receptor (IR) may elicit non-classic, ligand independent and tyrosine-kinase independent events that include reduction of apoptosis resistance, changes in DNA methylation and in imprinted genes and miRNAs. In the present paper they aim to identify the mechanisms underlying these effects and their relationship to receptor structure. To address these issues, they have generated mouse preadipocytes lacking both IR and IGF1R (DKO cells). DKO cells have then been reconstituted with: a) the B isoform of the human IR (hIR); b) a kinase dead hIR variant (K1030R cells); c) the hIR lacking the 79 amino acids of the C- terminus (Δ CT cells); d) a truncated hIR containing only the 36 juxtamembrane intracellular amino acids (JMO cells). All these cell lines, unstimulated or stimulated with 100mM of insulin for 15min, were then subjected to phosphoproteomics, gene expression and protein expression analysis.

They found that hIR mediates ligand independent and tyrosine-kinase independent (LYK-I) effects that can be reproduced by the K1030R variant and by the Δ CT IR but not by the JMO domain.

Relevant LYK-I included changes in phosphorylation of proteins involved in the regulation of extracellular matrix organization, cell cycle and ATM signaling along with upregulation of extracellular matrix-related genes and proteins and down-regulation of immune/interferon-related genes and proteins.

While LYK-I effects have already reported by the same authors in the same cell model, the present paper adds a significant body of new data that are likely to be of considerable interest for researchers in the field.

Comments

- 1) A limitation of the present study is that it does not fully elucidate the molecular mechanism(s) involved in LIK-I changes of protein phosphorylation and gene expression.
- 2) Other limitations of the article include the fact that the results are representative of a single cell model (mouse preadipocytes) and include only one (isoform B) of the two hIR isoforms. Although it is understandable that the generation of similar cell models is quite complex, these limitations should be clearly stated.
- 3) The gene expression analysis seems to have been mainly aimed at the description of LIK-I effects. However, it would have been useful to include insulin stimulation time points at 3-4h and/or 6-8h. This would have allowed to compare gene expression in DKO cells not only with LIK-I gene expression but also with classical insulin-stimulated gene expression. As also noted by the authors, only few

genes are stimulated by insulin at 15 min.

4) The authors may wish to speculate regarding the physiological meaning of these LIK-I effects.

5) In Fig. 1C the nuclear fraction of IR seems to be comparable to the cytoplasm fraction and higher than expected also by looking at the membrane fraction. Noteworthy the cells are unstimulated. Does the IR nuclear fraction increase after insulin stimulation?

6) Can the author quantify IR expression in reconstituted DKO cells? Is IR overexpressed or comparable to wt preadipocytes?

7) Experimental Procedures: DKO cells are compared with cells transduced with retroviral expression vectors encoding either wild-type or mutant IRs. It is unclear these DKO cells were transduced with the empty vector, which would be the proper control.

8) Fig. 5.: Please show cell confluency of the different cell lines as it is known that the expression of ISGs is dependent on cell confluency.

9) In contrast to Fig. 6, in supplementary Fig. 7b the expression of STAT3 in DKO cells is similar to that of IR-expressing cells, especially in serum-free conditions.

10) DKO cells are more resistant to apoptosis than IR containing cells, yet they express higher levels of ISGs that are usually pro-apoptotic. Please comment.

11) hIR expressing cells were characterized by increased levels of SASP related genes but lower levels of ISGs as compared to DKO cells. However, there is evidence that senescent cells show constitutive expression of interferon-induced genes (doi:10.1038/s41514-018-0030-6), which seems in contrast with these findings. Please comment.

Minor points: Page 19, Discussion: '...conical signaling pathway' should probably read '...canonical signaling pathway'

Reviewer #3:

Remarks to the Author:

The now widely accepted concept of insulin action (in which the senior author has played a major role) postulates that classical metabolic and mitogenic actions of insulin are mediated by insulin binding to its cell membrane receptor and activating its intracellular tyrosine kinase, resulting in receptor autophosphorylation and phosphorylation of intracellular signaling proteins. In this exciting and provocative paper, the authors explore the possibility that some biological effects of insulin mediated through the insulin receptor may in fact be ligand and tyrosine kinase-independent, as some of their data had previously suggested (refs 13 and 14). They use a clever strategy to investigate this issue. In order to define receptor-dependent but ligand- and kinase-independent signaling, they generated preadipocytes in which the endogenous IR and IGF1R have been genetically inactivated to create DKO cells. Then they reconstituted the cells with either the normal wild-type IR, a kinase-dead mutant IR (K1030R), a truncated IR lacking 79 amino acids from the C-terminus (Δ CT), or a truncated IR lacking almost the entire intracellular domain including both tyrosine-kinase domain and C-terminus, leaving only the intracellular juxtamembrane 36 amino acids (Juxtamembrane-Domain-Only, JMO). They then performed comprehensive molecular phenotyping of these variants in the absence and presence of insulin, including global phosphoproteomics, gene expression and protein expression analysis to understand the domain-dependent roles of the occupied and unoccupied receptors. They show conclusively that, in addition to the classical ligand and tyrosine kinase-dependent (LYK-D) signaling, the IR, even in the unliganded state, modulates a network of protein phosphorylation linked to control of expression of genes and proteins in several key regulatory pathways. These events (LYK-I) are largely dependent on the presence of the intracellular domain of the IR, but independent of receptor kinase activity or ligand binding, and include a major remodeling of the extracellular matrix, expression of cell cycle and immune-related genes and proteins, and a senescence-associated phenotype. These ligand- and tyrosine kinase-independent (LYK-I) signals also regulate sensitivity to apoptosis by both intrinsic and extrinsic pathways. These data suggest that a second, novel signaling state of the IR which is independent of ligand occupancy or kinase activity, based on the presence of a domain of the β -subunit of the IR between residues 989 and 1276 (although the JMO has effects of its own), may play unique roles in biology and pathophysiology and provides new approaches to treatment of diseases in which these pathways are altered.

This study may be a game-changer in our understanding of insulin actions and opens a wide new field of investigations. The paper is very thorough, quite dense and complex, copiously illustrated, and requires sustained attention, but is altogether very logical and well constructed. Fig. 8e nicely clarifies

everything.

I have only minor comments and suggestions, mostly to increase clarity.

General comments

The paper contains numerous abbreviations, a list of abbreviations would be helpful.

I have a problem with the way the schematic structure of the IR is shown in Figs 1a and 8e. Such a T-shaped symmetrical structure of the ectodomain, and with the intracellular kinase domains juxtaposed, is only observed in ligand saturated and kinase activated receptors (see Lawrence MC, Mol. Metab. 2021 for a review of IR recent structure data, although there are even more recent papers). The unliganded ectodomain has a symmetrical inverted V shape with the membrane inserts 120 angstroms apart. The partially liganded receptor has an asymmetrical ectodomain. I realize that you probably do not intend this scheme to be anatomically correct, but it is somewhat misleading and may suggest that you are not aware of the recent structural data. You may want to rethink this, maybe just showing it liganded.

I wonder if the data preclude the existence of ligand-dependent but kinase-independent IR effects, as suggested by Ken Siddle's and Ira Goldfine's groups in 1989 for insulin-like metabolic effects induced by IR monoclonal antibodies (Soos MA et al., PNAS USA 86, 5217-5221; Hawley, DM., et al., JBC 264, 2438-2444).

Specific comments

Page 4, last line: ...domain-dependent... (hyphen).

Page 5, last line: kinase activity that plays ...: delete "that".

Page 7 lines 6-7: ... a functional kinase domain... is crucial for... cell proliferation...: the K1030R mutant IR shows significantly increased cell proliferation (Fig. 1f).

Page 7 line 12: ... stimulated with 100 nM insulin stimulation...: delete "stimulation".

Page 8 line 9: ... up-regulated cells...: should be "in cells".

Page 8 lines 16-17: ... enrichment in... UPF1...: supplementary Fig. 2c shows a marked decrease in UPF1 in insulin-stimulated Δ CT cells.

Page 8 last 3 lines, page 9 first 4 lines: when you mentioned increased or low, you should specify "compared to DKO cells".

Page 9 line 2: define IR-ICD.

Page 10 line 18 and elsewhere: receptor isoforms: I would not use the term isoforms, which usually defines naturally occurring similar proteins that originate from a single gene and differ due to alternative splicing, variable protomer usage or other post-transcriptional modifications. Receptor variants or mutated receptors is better, as you also use elsewhere.

Page 11, line 17: differentially regulated up- or down- regulated: delete the first "regulated".

Page 13, lines 4-7: sentence somewhat unclear, should be rephrased.

Page 19, line 3: canonical not conical.

Page 20 line 3-4: ... are also almost all ligand independent: it means that some are still ligand-dependent (see my question above)?

Page 21, line 22: resistant, not resistance.

Page 23, line 13: ... this novel signaling pathways: "novel" and "pathways" both singular or plural.

References

Some references are not complete (see PubMed):

Ref 4: should be 6 (1): a009191 (2014)

Ref. 5: should be: 131 (10): e142241 (2021).

Ref. 6: should be: 43: 314-328 (2021).

Ref. 10: should be: 118: (17): e2109474118 (2021).

Figure 2c

The area marked "complex" is not defined or discussed.

Pierre De Meyts, MD, PhD.

Reviewer #4:

Remarks to the Author:

Nagao and colleagues have performed an investigation of the signaling induced by activation of the insulin receptor with emphasis on kinase-independent regulation. They report receptor involvement in cell cycle, senescence and apoptosis independent of the kinase activity of the receptor. They utilize

different constructs of the insulin receptor expressed in brown preadipocytes in which the endogenous insulin and the IGF1 receptors have been genetically inactivated. They report signaling events dependent on the intracellular domain, but independent on ligand binding and independent on tyrosine kinase activity. The authors primarily evaluate the signaling outcome based on proteomics and phosphoproteomics experiments. Overall, the authors have performed an interesting study. However, the study is largely descriptive in nature. Further investigations of the biological relevance of their findings and the functional implications would strengthen the study.

Major critiques:

1) The authors state that DKO cells lack both IR and IGF1R, and they show this by western blot in Figure 1. However, this is not recapitulated in the proteomics dataset they provided, where both receptors are reported with a protein intensity greater than 1e9 in the three DKO replicates. Please clarify?

2) In Figure 1D the authors show a very convincing western blot showing phosphorylation of Akt S473 upon insulin stimulation in the IR calls and in the delta CT cells. However, these results do not seem to be in line with the data reported in the phosphoproteomics dataset: only one out of three IR replicates have a higher phosphorylation level of AktS473 upon insulin stimulation. On the contrary, the phosphoproteomics data show higher phosphorylation levels in the kinase dead construct. There appears to be a discrepancy between the representative western blot and the phosphoproteomics data. Please clarify.

3) Overall, there is not sufficient details provided for the omics experiments and analyses. This is deemed critical as all conclusions are based on these datasets. Please:

-expand on the experimental procedure instead of referring to two previous papers

-add details on how the data processing was done

-add details on all analyses performed

-specific point on data imputation: In one of the two papers the authors refer to for how they conducted their analyses they write in the methods section "... phosphopeptides that have at least two valid values were selected, and their missing values were imputed with half of the minimum value of the corresponding phosphopeptide...". If the authors have utilized a similar approach here they need to explain this in detail. And they need to justify their approach, especially if their imputation strategy is not applied across the full dataset.

-add specifics of pathway analyses

-there is a lack of explanation of the content of the supplementary datasets. Please add details on the content. Preferentially details on the content of each column.

4) The authors do not include information on differentially regulated proteins and phosphorylated peptides. This should be added to the Excel files.

5) In the Supplementary Dataset with phosphoproteomics data the authors lack critical information. For a lot of reported phosphorylation sites the authors provide several lines of data regarding measured intensities for that particular phosphorylation site. Surely this is not correct. As example, for Irs1 S1211 the authors first report one set of intensities where it seems the phosphorylation levels are more abundant in the DKO basal cells than in the insulin stimulated IR cells. Then they report a second set of intensities where the intensities are greater in the insulin stimulated IR cells than in the DKO cells. I will have to assume that the reason of this apparent discrepancy has to do with multi-phosphorylated peptides. The authors are simplifying the dataset with their representation in an unjustifiable manner. Information on multiplicity should be included.

6) Please specify how the KSEA analysis was performed. Was the analysis restricted to mono-phosphorylated peptides? How were the data transformed to human orthologs? Please also provide the actual data from the analysis.

7) Can the authors please explain how they have performed statistical evaluation across conditions with only two replicates (delta CT), as in e.g. (but not limited to) Fig.2b, Fig2.e, Fig.3a etc.

8) In general, how do the authors justify their statistical evaluation of particular phosphorylation sites and proteins? Just as one example (but again not limited to), the authors report on "Quantitation of exemplary phosphosites" in Figure 3A. These intensities are extracted from a dataset where the authors evaluated close to 15,000 peptides. Yet they report means +/- SEM using one-way ANOVA. Can the authors please justify this approach considering multiple hypothesis testing.

9) For interpretation of the data presented in Figure 2, it would be useful to also see data from DKO cells stimulated with insulin.

REVIEWER COMMENTS

Reviewer #1 (Remarks to the Author):

This manuscript uncovers a novel regulatory role of the insulin receptor intracellular domain. This tour de force omics study combines phosphoproteomics, RNAseq and proteomics to uncover novel functions of the insulin receptor beyond its normal ligand-kinase functions. While this work has the potential to be very exciting, as it suggests a new role of the insulin receptor in apoptosis and senescence, some concerns should be met before it can be considered for publication.

Notably, this entire work was conducted only utilizing a single cell line that was isolated from BAT and immortalized more than 12 years ago. While it was key to utilize the double knockout background for initial studies, some of the main effects of the paper need to be validated in other cell lines to demonstrate that these findings are generalizable beyond this one unique cell line. Thus, it will be helpful to repeat some of the main findings using re-expression of mouse insulin receptor as these effects may result from expressing a mutant human protein into a mouse cell line.

We thank the reviewer for the valuable comment. Based on the advice, we validated some of the main findings of LYK-I actions by using an independent set of DKO preadipocytes reconstituted with the mouse insulin receptor (mIR) rather than the human IR. As in the human IR re-expressing cells, re-expression of mIR also rescued apoptotic sensitivity, increased mRNA levels of collagens, and decreased mRNA levels of interferon and interferon stimulated genes. These results are shown in the **new Supplementary Fig. 8.**

Supplementary Fig. 8. Changes in apoptotic sensitivity and gene expressions in DKO and mouse IR cells.

(a) Immunoblot analysis of WT preadipocytes, DKO preadipocytes and DKO preadipocytes reconstituted with mouse IR (mIR, A isoform) cells for IR β , IGF1R β and β -actin, and quantification of IR β . Data are means \pm SEM (n = 3 per group). *** P < 0.001. (b) Immunoblotting of IRS-1^{Y608} phosphorylation and Akt^{S473} phosphorylation in lysates from DKO and mIR cells stimulated with 100 nM insulin for 15 min. (c) Immunoblotting of cleaved caspase 3 in DKO and mIR cells following FBS starvation for 6h. For densitometric analysis of cleaved caspase 3, the level in mIR cells without FBS starvation was set at 1. (d) mRNA levels of collagens and genes associated with interferon signaling by qPCR.

A large portion of the discussion restates the results section, instead the authors may wish to discuss why they believe the insulin receptor would evolve these alternative functions

and importantly promote apoptosis?

We really appreciated this suggestion. Based on this, we removed some sentences which restate the results section and added some sentences about why the insulin receptor would promote apoptosis in the un-liganded state in the **DISCUSSION** section (Page 23).

All graphs should show individual data points. Specially figure 8 and figure 1.

We agree, and based on this advice and the journal policies, we have added individual data points in all graphs.

Reviewer #2 (Remarks to the Author):

The authors have previously reported that the Insulin Receptor (IR) may elicit non-classic, ligand independent and tyrosine-kinase independent events that include reduction of apoptosis resistance, changes in DNA methylation and in imprinted genes and miRNAs. In the present paper they aim to identify the mechanisms underlying these effects and their relationship to receptor structure. To address these issues, they have generated mouse preadipocytes lacking both IR and IGF1R (DKO cells). DKO cells have then been reconstituted with: a) the B isoform of the human IR (hIR); b) a kinase dead hIR variant (K1030R cells); c) the hIR lacking the 79 amino acids of the C- terminus (DCT cells); d) a truncated hIR containing only the 36 juxtamembrane intracellular amino acids (JMO cells). All these cell lines, unstimulated or stimulated with 100mM of insulin for 15min, were then subjected to phosphoproteomics, gene expression and protein expression analysis.

They found that hIR mediates ligand independent and tyrosine-kinase independent (LYK-I) effects that can be reproduced by the K1030R variant and by the DCT IR but not by the JMO domain.

Relevant LYK-I included changes in phosphorylation of proteins involved in the regulation of extracellular matrix organization, cell cycle and ATM signaling along with upregulation of extracellular matrix-related genes and proteins and down-regulation of immune/interferon-related genes and proteins.

While LYK-I effects have already reported by the same authors in the same cell model, the present paper adds a significant body of new data that are likely to be of considerable interest for researchers in the field.

Comments

1) A limitation of the present study is that it does not fully elucidate the molecular mechanism(s) involved in LYK-I changes of protein phosphorylation and gene expression.

We agree with the reviewer's comment that we haven't fully elucidated the molecular mechanisms of LYK-I actions and believe that this will require many additional experiments over the next 2-3 years. We do believe, however, that we have made an important new observation and made significant progress toward defining the structural features of the IR that are involved in LYK-I signaling. However, based on the reviewer's comment, in the revised manuscript we now discuss several potential mechanisms that could account for these observations based on our dataset. One potential mechanism is

effects of the receptor in different compartments of the cells as it migrates through the cell where it might interact with other proteins in the cell in a kinase-independent fashion that create some novel secondary signaling, equivalent to the role of the beta-arrestins in G-protein coupled receptor signaling. Another potential mechanism of LYK-I effects could be direct transcriptional control by the IR or some fragments of the IR in nucleus as suggested by the work of Hancock et al (Cell 2019). Rest assured that no one is more anxious than us to identify the exact molecular mechanism(s) underlying the LYK-I actions of the insulin receptor.

2) Other limitations of the article include the fact that the results are representative of a single cell model (mouse preadipocytes) and include only one (isoform B) of the two hIR isoforms. Although it is understandable that the generation of similar cell models is quite complex, these limitations should be clearly stated.

We thank the reviewer for the valuable suggestion. As noted in the response to Reviewer 1, we have now confirmed some of the key findings in a new set of DKO cells transfected with the murine IR A isoform. Nonetheless, based on this advice, we have added this limitation to the **DISCUSSION** sections (Page 25) as follows.

“At present, we have studied these effects only in a single cell model (mouse preadipocytes), so further studies will be needed to determine if these LYK-I are similar in different cell types, or whether different cell types have a unique program of LYK-I effects.”

3) The gene expression analysis seems to have been mainly aimed at the description of LIK-I effects. However, it would have been useful to include insulin stimulation time points at 3-4h and/or 6-8h. This would have allowed to compare gene expression in DKO cells not only with LIK-I gene expression but also with classical insulin-stimulated gene expression. As also noted by the authors, only few genes are stimulated by insulin at 15 min.

We agree with the reviewer that it would have been interesting to study the changes in gene expression at multiple time point; however, for RNAseq we concentrated on cells in the basal state and after short stimulation which are the early immediate response genes. However, using qPCR we have analyzed changes in genes expression in these cells following longer term insulin stimulation (100 nM insulin for 6 h) in DKO cells not only

on genes altered by through LYK-I actions, but also those associated with classical insulin-stimulated gene expression. These new findings have been added as **new Fig.1g and new Supplementary Fig. 4b-f** and are discussed in the Results section on pages 7 and 13 as follows.

(Page 7)

Likewise, mRNA levels of phosphofruktokinase, liver type (*Pfkl*) were significantly up-regulated by 6 h insulin stimulation only in cells expressing the intact IR and Δ CT mutants (Fig. 1g).

(Page 13)

We also analyzed changes in mRNA expression of selected genes in these cell lines following long term insulin stimulation (100 nM insulin for 6 h) by qPCR (Supplementary Fig. 4b-f). Well-known insulin regulated genes, such as *Srebp1c* and *Pfkl*, were significantly up-regulated by insulin only in IR and Δ CT cells (Supplementary Fig. 4b and Fig. 1g). mRNAs for genes in the “High Expression in IR-ICD” cluster, i.e., collagens and cholesterol synthesis related genes, such as *Fdps* and *Sqle*, were slightly up-regulated in IR and Δ CT cells by insulin (Supplementary Fig. 4c,d), while mRNA levels of interferons and IFN-ISGs, representing genes in the “Low Expression in IR-ICD” cluster, showed little change following insulin stimulation (Supplementary Fig. 4e,f).

4) The authors may wish to speculate regarding the physiological meaning of these LIK-I effects.

We appreciate this suggestion and have added some sentences about the physiological meaning of these LYK-I effects in the **DISCUSSION** section on pages 23.

5) In Fig. 1C the nuclear fraction of IR seems to be comparable to the cytoplasm fraction and higher than expected also by looking at the membrane fraction. Noteworthy the cells are unstimulated. Does the IR nuclear fraction increase after insulin stimulation?

Based on the comment, we analyzed the localization of these wild-type IR and mutated receptors following insulin stimulation. At least over 30 min, insulin stimulation (100 nM) did not increase localization of the fraction of wild-type IR and mutated receptors appearing in the nucleus fraction. These new data are shown in **new Supplementary Fig. 1c**, and we have added a following sentence in the **Results** section (Page 6).

“A small fraction of each of these receptors could also be identified in extracts of the nuclear fraction, and these did not change with 30 min of insulin stimulation (Supplementary Fig. 1c).”

Supplementary Fig. 1c. Immunoblotting of IR β in lysates of membrane (Mem) and nucleus (Nuc) fraction from DKO, IR, K1030R, Δ CT and JMO cells. Cells were serum starved 6 h and stimulated with 100 nM insulin for 30 min or left untreated.

6) Can the author quantify IR expression in reconstituted DKO cells? Is IR overexpressed or comparable to wt preadipocytes?

Exact comparison of the various IR constructs was estimated both at the mRNA and protein level. Figure 1B shows the mRNA levels of the different receptor constructs and the level of endogenous IR expression in wild-type cells. As noted in the figure legend, the “Dotted line represents the level of IR mRNA in WT cells.” While the level of mRNA expression of the recombinant receptors in these cell lines was about four times the level of endogenous IR in the wild-type preadipocytes, it is not at the very high levels used in many transfection experiments.

Based on this, we added a following sentence on Page 6.

“On average the mRNA levels of the exogenous receptor constructs expressions were about four times higher than normal endogenous IR in wild-type (WT) cells, indicating only a mild degree of overexpression.”

The level of receptor protein as determined by immunoblotting of IR β in these cells paralleled that of mRNA, although there is the caveat that the different beta-subunits may

have slightly different conformations resulting in different reactivity with insulin receptor antibodies, and the fact that lower molecular weight proteins may transfer better in western blotting experiment, with the recombinant receptors being 2-4 fold higher than the normal endogenous levels. This is shown in figure below and in Figure 1d in the manuscript.

7) Experimental Procedures: DKO cells are compared with cells transduced with retroviral expression vectors encoding either wild-type or mutant IRs. It is unclear these DKO cells were transduced with the empty vector, which would be the proper control.

We thank the reviewer for pointing this out. Based on the comment, we revised this section (Page 27) in the **Experimental Procedures** to properly describe the control cells.

8) Fig. 5.: Please show cell confluency of the different cell lines as it is known that the expression of ISGs is dependent on cell confluency.

To address the reviewer's question, we have added the photographic images of DKO, IR, K1030R, Δ CT and JMO cells (new **Supplementary Fig. 2a**). We used these confluent cells for experiments. Therefore, we also added this information to the Result section of the manuscript (Page 7).

9) In contrast to Fig. 6, in supplementary Fig. 7b the expression of STAT3 in DKO cells is similar to that of IR-expressing cells, especially in serum-free conditions.

Based on the comment, we have analyzed the densities of STAT3 bands in Supplementary Fig. 7b (shown below for the reviewer). In this analysis, protein levels of STAT3 in IR cells were decreased 10-20% compared to those in DKO cells. However, STAT3 levels in IR, K1030R and Δ CT cells were decreased 25-30% compared to DKO cells in Fig. 6h.

Thus, although the effect was smaller, the same trend was observed.

Figure for the reviewer.

10) DKO cells are more resistant to apoptosis than IR containing cells, yet they express higher levels of ISGs that are usually pro-apoptotic. Please comment.

We thank the reviewer for the comment. DKO cells (and JMO cells) are more resistant to apoptosis than IR, K1030R and Δ CT cells. Resistance to apoptosis is one of the hallmarks of cellular senescence. Senescent cells also show up-regulation of interferons, interferon-stimulated genes (ISGs) and SASP related genes. These are also observed in DKO cells (Fig.5). Thus, DKO cells (and JMO cells) showed increased cellular senescence phenotypes. However, as the reviewer mentioned, higher levels of ISGs usually act as mediators of apoptosis, which seems to be conflicting with the phenotype of cellular senescence. While this complex phenotype will need further study, it could be compensatory change in apoptosis resistance.

Together with the comment 11 below, we have added a sentence to the **DISCUSSION** section (Page 22) as follows.

“Higher expression of interferons, ISGs and SASP related genes, which are observed in DKO and JMO cells, are sometimes regarded as hallmarks of senescent cells (43, 44).”

11) hIR expressing cells were characterized by increased levels of SASP related genes but lower levels of ISGs as compared to DKO cells. However, there is evidence that senescent cells show constitutive expression of interferon-induced genes (doi:10.1038/s41514-018-0030-6), which seems in contrast with these findings. Please comment.

Our dataset showed decreased SASP related genes in parallel with decreased ISGs in hIR expressing cells compared to DKO cells (Fig. 5d,e,f). We also showed that hIR cells (also K1030R and Δ CT cells) showed decreases in some aspects of the senescence phenotype (Fig. 8e). Therefore, we think these findings consistent with it.

Minor points: Page 19, Discussion: ‘...conical signaling pathway’ should probably read ‘...canonical signaling pathway’

We thank the reviewer for pointing it out. We revised this typo.

Reviewer #3 (Remarks to the Author):

The now widely accepted concept of insulin action (in which the senior author has played a major role) postulates that classical metabolic and mitogenic actions of insulin are mediated by insulin binding to its cell membrane receptor and activating its intracellular tyrosine kinase, resulting in receptor autophosphorylation and phosphorylation of intracellular signaling proteins. In this exciting and provocative paper, the authors explore the possibility that some biological effects of insulin mediated through the insulin receptor may in fact be ligand and tyrosine kinase-independent, as some of their data had previously suggested (refs 13 and 14). They use a clever strategy to investigate this issue. In order to define receptor-dependent but ligand- and kinase-independent signaling, they generated preadipocytes in which the endogenous IR and IGF1R have been genetically inactivated to create DKO cells. Then they reconstituted the cells with either the normal wild-type IR, a kinase-dead mutant IR (K1030R), a truncated IR lacking 79 amino acids from the C-terminus (DCT), or a truncated IR lacking almost the entire intracellular domain including both tyrosine-kinase domain and C-terminus, leaving only the intracellular juxtamembrane 36 amino acids (Juxtamembrane-Domain-Only, JMO). They then performed comprehensive molecular phenotyping of these variants in the absence and presence of insulin, including global phosphoproteomics, gene expression and protein expression analysis to understand the domain-dependent roles of the occupied and unoccupied receptors. They show conclusively that, in addition to the classical ligand and tyrosine kinase-dependent (LYK-D) signaling, the IR, even in the unliganded state, modulates a network of protein phosphorylation linked to control of expression of genes and proteins in several key regulatory pathways. These events (LYK-I) are largely dependent on the presence of the intracellular domain of the IR, but independent of receptor kinase activity or ligand binding, and include a major remodeling of the extracellular matrix, expression of cell cycle and immune-related genes and proteins, and a senescence-associated phenotype. These ligand- and tyrosine kinase-independent (LYK-I) signals also regulate sensitivity to apoptosis by both intrinsic and extrinsic pathways. These data suggest that a second, novel signaling state of the IR which is independent of ligand occupancy or kinase activity, based on the presence of a domain of the b-subunit of the IR between residues 989 and 1276 (although the JMO has effects of its own), may play unique roles in biology and pathophysiology and provides new approaches to treatment of diseases in which these pathways are altered. This study may be a game-changer in our understanding of insulin actions and opens a wide new field of investigations. The paper is very thorough, quite dense and complex,

copiously illustrated, and requires sustained attention, but is altogether very logical and well constructed. Fig. 8e nicely clarifies everything.

I have only minor comments and suggestions, mostly to increase clarity.

General comments

The paper contains numerous abbreviations, a list of abbreviations would be helpful.

I have a problem with the way the schematic structure of the IR is shown in Figs 1a and 8e. Such a T-shaped symmetrical structure of the ectodomain, and with the intracellular kinase domains juxtaposed, is only observed in ligand saturated and kinase activated receptors (see Lawrence MC, Mol. Metab. 2021 for a review of IR recent structure data, although there are even more recent papers). The unliganded ectodomain has a symmetrical inverted V shape with the membrane inserts 120 angstroms apart. The partially liganded receptor has an asymmetrical ectodomain. I realize that you probably do not intend this scheme to be anatomically correct, but it is somewhat misleading and may suggest that you are not aware of the recent structural data. You may want to rethink this, maybe just showing it liganded.

I wonder if the data preclude the existence of ligand-dependent but kinase-independent IR effects, as suggested by Ken Siddle's and Ira Goldfine's groups in 1989 for insulin-like metabolic effects induced by IR monoclonal antibodies (Soos MA et al., PNAS USA 86, 5217-5221; Hawley, DM., et al., JBC 264, 2438-2444).

We thank the reviewer for the valuable comment and suggestion. Based on the advice, the schematic structure of the IRs in Figs 1a and 8e were added the schema of ligands. We have also included a list of abbreviations in the supplement.

Specific comments

We really appreciate these comments. We revised the manuscript properly based on these points.

Page 4, last line: ...domain-dependent... (hyphen).

Revised as suggested.

Page 5, last line: kinase activity that plays ...: delete "that".

Revised as suggested.

Page 7 lines 6-7: ... a functional kinase domain... is crucial for... cell proliferation...: the K1030R mutant IR shows significantly increased cell proliferation (Fig. 1f).

Based on this, we revised the **Results** section (page 7) as follows:

“Together, these data demonstrate that the intracellular domain of the IR, including a functional kinase domain but not the C-terminal 79 amino acids, is critical for ligand-stimulated glycolysis and adipocyte differentiation. Cells expressing the K1030R mutant also showed significantly higher cell proliferation than DKO cells, but not to the extent of cells with a kinase active receptor. Cells expressing IR-JMO also may have some unique effects, as illustrated by the effect to reduce glycolysis.”

Page 7 line 12: ... stimulated with 100 nM insulin stimulation...: delete “stimulation”. This has been revised as suggested.

Page 8 line 9: ... up-regulated cells...: should be “in cells”. This has been revised as suggested.

Page 8 lines 16-17: ... enrichment in... UPF1...: supplementary Fig. 2c shows a marked decrease in UPF1 in insulin-stimulated DCT cells. Based on the comment, we revised the text as follows.

“Pathway analysis of this “ Δ CT specific” cluster revealed that it included proteins involved in cell cycle and RNA transport. For example, BRCA1^{S717}, a protein involved in mRNA nuclear export, was up-regulated by insulin, and UPF1^{S1102}, a protein involved in mRNA surveillance and initiation of nonsense-mediated mRNA decay, was down-regulated by insulin in a manner dependent on the receptor C-terminal domain”

Page 8 last 3 lines, page 9 first 4 lines: when you mentioned increased or low, you should specify “compared to DKO cells”. As requested by the reviewer, we have now specified “compared to DKO cells” in these sentences.

Page 9 line 2: define IR-ICD.

The insulin receptor intracellular domain is now defined as IR_ICD.

Page 10 line 18 and elsewhere: receptor isoforms: I would not use the term isoforms, which usually defines naturally occurring similar proteins that originate from a single gene and differ due to alternative splicing, variable promoter usage or other post-transcriptional modifications. Receptor variants or mutated receptors is better, as you also use elsewhere.

We thank the reviewer for the valuable comment. We stopped using the term “isoforms” and used appropriate terms such as “mutated receptors” in the whole manuscript.

Page 11, line 17: differentially regulated up- or down- regulated: delete the first “regulated”.

We deleted the first “regulated”.

Page 13, lines 4-7: sentence somewhat unclear, should be rephrased.

We thank the reviewer for the comment. To make this sentence clearer, we rephrased it as follows.

“A limited correlation between mRNA and protein levels is a well-known biological phenomenon (33), but to what extent this is modulated by metabolic cues is less well understood.”

Page 19, line 3: canonical not conical.

We thank the reviewer for pointing out this typo. We have revised it.

Page 20 line 3-4: ... are also almost all ligand independent: it means that some are still ligand-dependent (see my question above)?

We apologize for the inaccurate representation. These phosphorylation events are entirely ligand independent. Based on this, we revised this sentence as “these phosphorylation events are also ligand independent,”

Page 21, line 22: resistant, not resistance.

We thank the reviewer for pointing out the typo. It has been revised.

Page 23, line 13: ... this novel signaling pathways: “novel” and “pathways” both singular

or plural.

We thank the reviewer for pointing it out. We revised the sentence to say “these...”.

References

Some references are not complete (see PubMed):

Ref 4: should be 6 (1): a009191 (2014)

Ref. 5: should be: 131 (10): e142241 (2021).

Ref. 6: should be: 43: 314-328 (2021).

Ref. 10: should be: 118: (17): e2109474118 (2021).

Thank you for these comments on references. We used the software “EndNote” for referencing, which is supposed to be updated to the correct reference style for each journal. We have rerun the manuscript through EndNote’s with the most updated file for Nature Communications, so hopefully all references are now correct.

Figure 2c

The area marked “complex” is not defined or discussed.

Based on the suggestion, we revised that the area marked “complex” is not defined.

Pierre De Meyts, MD, PhD.

Reviewer #4 (Remarks to the Author):

Nagao and colleagues have performed an investigation of the signaling induced by activation of the insulin receptor with emphasis on kinase-independent regulation. They report receptor involvement in cell cycle, senescence and apoptosis independent of the kinase activity of the receptor. They utilize different constructs of the insulin receptor expressed in brown preadipocytes in which the endogenous insulin and the IGF1 receptors have been genetically inactivated. They report signaling events dependent on the intracellular domain, but independent on ligand binding and independent on tyrosine kinase activity. The authors primarily evaluate the signaling outcome based on proteomics and phosphoproteomics experiments. Overall, the authors have performed an interesting study. However, the study is largely descriptive in nature. Further investigations of the biological relevance of their findings and the functional implications would strengthen the study.

Major critiques:

1) The authors state that DKO cells lack both IR and IGF1R, and they show this by western blot in Figure 1. However, this is not recapitulated in the proteomics dataset they provided, where both receptors are reported with a protein intensity greater than $1e9$ in the three DKO replicates. Please clarify?

We thank the reviewer for the valuable comment.

Briefly, we choose data-dependent acquisition (DDA) mode to measure the proteome, which allows capture of the difference more clearly between WT vs KO samples. Regarding the INSR expression, it was neither detected nor quantified at both peptide and protein levels in the DKO samples (see **new Supplementary Table 1** which displays all INSR peptide identified in the dataset). The quantified protein intensity presented in the Excel table does give “values” for INSR. This is because these data were listed after filtering sample group, which gives a value if 80% of samples were present in at least in one group. This is the default procedure for this type of analysis in proteomics. The analysis was then followed by a data imputation step (assuming normal Gaussian distribution (width 0.5 and downshift 1.5)) based on the assumption that data missing were not by random (MNBR). It was this data filtering strategy which included filtering by groups that resulted in the suggestion that the INSR was still present in the DKO cells since other samples groups expressed this protein. Therefore, we continue to show these imputed data points although they are nearly 8-fold lower in DKO cells compared to other

samples ,even after data imputation (in the Figure as follows), although this is essentially background. On the other hand, even with this method, we confirmed that IGF1R was not detected in the proteomic dataset.

Figure for the reviewer

2) In Figure 1D the authors show a very convincing western blot showing phosphorylation of Akt S473 upon insulin stimulation in the IR calls and in the delta CT cells. However, these results do not seem to be in line with the data reported in the phosphoproteomics dataset: only one out of three IR replicates have a higher phosphorylation level of AktS473 upon insulin stimulation. On the contrary, the phosphoproteomics data show higher phosphorylation levels in the kinase dead construct. There appears to be a discrepancy between the representative western blot and the phosphoproteomics data. Please clarify.

As the reviewer pointed it out, Akt S473 in the phosphoproteomic dataset appears to be a different from the representative western blot data. In the Western blot there was clear stimulation by both the WT IR and the delta CT-IR, whereas in phosphoproteomics, only the WT-IR showed significant stimulation. This may reflect differences in which isoforms of Akt are measured in each assay. Note that the phospho-Akt (S473) antibody used in these studies (#4060, Cell Signaling) recognizes Akt2 (S474) and Akt3 (S472) in addition to Akt1 at Ser473. And as shown in Figure 2E below, phosphorylation of Akt3 (S472) was very similar to the representative western blot data of Akt phosphorylation. This likely explains the difference between the phosphoproteomics data and western blot data in Figure 1D.

Based on it, we added a following sentence in the **Results** section (Page 8).

(Page 8)

The phosphorylation of Akt3^{S472} was similar to the western blot data of Akt phosphorylation in Figure 1d. The phospho-Akt antibody recognizes Akt1 (S473), Akt2 (S474) and Akt3 (S472).

3) Overall, there is not sufficient details provided for the omics experiments and analyses. This is deemed critical as all conclusions are based on these datasets. Please:

-expand on the experimental procedure instead of referring to two previous papers

-add details on how the data processing was done

-add details on all analyses performed

-specific point on data imputation: In one of the two papers the authors refer to for how they conducted their analyses they write in the methods section "... phosphopeptides that have at least two valid values were selected, and their missing values were imputed with half of the minimum value of the corresponding phosphopeptide...". If the authors have utilized a similar approach here they need to explain this in detail. And they need to justify their approach, especially if their imputation strategy is not applied across the full dataset.

-add specifics of pathway analyses

-there is a lack of explanation of the content of the supplementary datasets. Please add details on the content. Preferentially details on the content of each column.

We apologize for the lack of some of the requested details. Based on these suggestions, we have expanded the experimental procedure section with details of how the data

processing was done, all analyses performed and the data imputation method (Page 31-33, “**Phosphoproteome and proteome analysis**” in the **Experimental Procedures** section and Page 28-29, “**Quantification of mRNA Levels by qPCR and RNA-sequencing**” in the **Experimental Procedures** section). We also added the explanation and the details of analyses including the sheets of moderated F-tests for the differentially regulated phosphosites, the hierarchical cluster analysis and the pathway analyses in the supplementary datasets.

4) The authors do not include information on differentially regulated proteins and phosphorylated peptides. This should be added to the Excel files.

We thank the reviewer for this suggestion. Based on this advice, we have added information on differentially regulated proteins and phosphorylated peptides to the Excel files.

5) In the Supplementary Dataset with phosphoproteomics data the authors lack critical information. For a lot of reported phosphorylation sites the authors provide several lines of data regarding measured intensities for that particular phosphorylation site. Surely this is not correct. As example, for Irs1 S1211 the authors first report one set of intensities where it seems the phosphorylation levels are more abundant in the DKO basal cells than in the insulin stimulated IR cells. Then they report a second set of intensities where the intensities are greater in the insulin stimulated IR cells than in the DKO cells. I will have to assume that the reason of this apparent discrepancy has to do with multi-phosphorylated peptides. The authors are simplifying the dataset with their representation in an unjustifiable manner. Information on multiplicity should be included.

We thank the reviewer for the comment. As the reviewer suggested, we included the multiplicity column for all the reported phosphosites in the phosphoproteome dataset excel sheet.

As we added in the **Experimental Procedures**, we worked with class-I phosphosites (with the localization cutoff set to >0.75). The heatmap analysis with differentially regulated phosphosites in the phosphoproteome uses all 15,000 phosphopeptides (unbiased analysis). As the reviewer pointed out, this dataset includes multiple phosphorylated peptides for the same phosphosite and it is the general nature of any phosphoproteome dataset which is due to ‘phosphorylation stoichiometry’. For example,

MAPK14 phosphorylation can occur in several multiplicity. MAPK14 T180 with multiplicity 1 represents the singly phosphorylated peptide, MAPK14 T180 with multiplicity 2 and MAPK14 Y182 with multiplicity 2 share the same intensity as they represent the two phosphosites on the same peptide. With the intensity of MAPK14 T180 with multiplicity 1 and MAPK14 T180 with multiplicity 2 differs as they are derived from two different peptides. For each phosphosite on a multiple phosphorylated peptide, we receive a row with the same intensities as these phosphorylations are localized on the same peptide. Different phosphosites on the same peptide can have slightly different fold changes due to imputation (after categorical grouping). However, in most cases, these phosphosites shows similar pattern between multiplicities. In the figure below, bar graphs of multiplicity in phosphopeptides are shown for some exemplary phosphosites presented in main Figures of the paper (this figure is for the reviewer only). These 4 sites have multiphosphorylated peptides in the exemplary phosphosites shown in Figure 2, 3 and 7. Based on it, we added a following sentence in the **DISCUSSION** section (Page 21).

(Page 21)

“It is worth noting that while the phosphoproteomic analysis includes multiple phosphopeptides in the same phosphosite, and that there was some variability as to the magnitude of the differences observed depending on the multiplicity state analyzed, many of the changes were quite robust”

Figures for the reviewer

6) Please specify how the KSEA analysis was performed. Was the analysis restricted to mono-phosphorylated peptides? How were the data transformed to human orthologs? Please also provide the actual data from the analysis.

We thank the reviewer for the comment. Based on the comment, we added detailed information of the kinase enrichment analysis in the **Experimental Procedures** section (Page 32). We also provided the actual data in the supplementary dataset. As we describe in the response to the comment 5, the analysis is not restricted to mono-phosphorylated peptides. This limitation has been added in the **Discussion** section as described in the response to the comment 5. The protein sets based on kinase substrates were curated specifically for mouse (details are in the website of PhosphositePlus, <https://www.phosphosite.org>) because we used mouse cell lines.

(Page 32)

Heatmaps were created with the pheatmap R package. Protein sets were tested to determine if the phosphosites in each cluster are over-represented. The protein sets based on kinase substrates (PhosphositePlus²⁴ and RegPhos²⁵) and phosphatases (the Dephosphorylation Database (DEPOD)⁷²) were tested using the Fisher exact in the R

software.

7) Can the authors please explain how they have performed statistical evaluation across conditions with only two replicates (delta CT), as in e.g. (but not limited to) Fig.2b, Fig2.e, Fig.3a etc.

While N=2 per group is actually sufficient to apply a t-test, N=2 is not recommended because it has low power. However, ANOVA estimates the variance of a gene across all groups, even in a test between two groups, which increases the power of the statistical analysis. We got p-values in Fig. 2b and 2e using two-way ANOVA followed by Šídák's multiple comparisons test. We understand two replicates is not ideal, and indeed for all sample we had n=3, except for the delta CT cells where one sample was lost and we had only n=2. In Figure 3A, we combined both the basal and the stimulated together, then compared between DKO and other cell lines using one-way ANOVA followed by Dunnett's multiple comparisons test. This information has been added in the Figure Legends.

8) In general, how do the authors justify their statistical evaluation of particular phosphorylation sites and proteins? Just as one example (but again not limited to), the authors report on “Quantitation of exemplary phosphosites” in Figure 3A. These intensities are extracted from a dataset where the authors evaluated close to 15,000 peptides. Yet they report means +/- SEM using one-way ANOVA. Can the authors please justify this approach considering multiple hypothesis testing.

We thank the reviewer for this comment. In Fig. 3a, these exemplary phosphosites were extracted from the significantly regulated phosphosites (ANOVA testing with stricter significance threshold, permutation based FDR cutoff <0.05) in the enriched REACTOME pathways of this “High-Phos in IR-ICD cluster” (example, in Fig. S3a). As with any examples, there were multiple sites to choose from, but we focused on those we found most interesting from a biological perspective. Furthermore, one-way ANOVA followed by Dunnett's multiple comparisons test or two-way ANOVA followed by Šídák's multiple comparisons test was performed and reported p-values hence adjusted for multiple comparison problems. We added this information in the Figure Legends. We also added information of how the phosphosites were extracted in Fig.S2e, S3b, S6d, S6f and 7a in the Figure Legends based on the comment. Of the significantly regulated proteins and phosphosites, when reported as a bar plot we also reported the +/- SEM as data points

were available for all the sample groups.

9) For interpretation of the data presented in Figure 2, it would be useful to also see data from DKO cells stimulated with insulin.

We thank the reviewer for the comment. Unfortunately, we don't have the data from DKO cells stimulated with insulin in this dataset, and it would be prohibitively expensive to repeat this whole experiment to add these data, since in the absence of INSR, we don't expect to see any phosphorylation change, as has been documented in previous papers from our lab using these cells (Boucher J et al. *Sci Signal*. 2010, Nagao H et al. *PNAS* 2021 and the current study new Supplementary Fig. 8b).

REVIEWER COMMENTS

Reviewer #2 (Remarks to the Author):

I have only two minor comments:

Supplementary Fig. 1c: The finding that the nuclear localization of IR is not increased after insulin exposure is apparently in contrast with the findings of Hancock et al. (Cell 177, 722–736, 2019) that showed increased chromatin-bound IR and IR interaction with transcriptional machinery in response to insulin stimulation. The hypothesis that: 'Another potential mechanism of LYK-I effects could be direct transcriptional control by the IR or some fragments of the IR in nucleus as suggested by the work of Hancock et al (Cell 2019)' seems also in contrast with Hancock's findings.

Supplementary Fig. 8: It is unclear why the Authors chose to reconstitute the mouse IR with the IR-A isoform while they used the human IR-B isoform in all the other experiments. The legend of Fig. 8 may contain a typo: '(a) Immunoblot analysis of WT preadipocytes, DKO preadipocytes and DKO preadipocytes reconstituted with mouse IR (mIR, A isoform) **cells** for IR β , IGF1R β and β -actin, and quantification of IR β .'

Reviewer #3 (Remarks to the Author):

I think the authors have satisfactorily met the reviewers critiques, it is certainly the case for mine.

Reviewer #4 (Remarks to the Author):

In this revision, the authors have provided answers to the questions raised. Yet, for the majority of points additional clarification is encouraged. Details pertaining the same questions as was put forward in the first revision is asked for below.

Response to point 1:

I would encourage the authors to improve the presentation of data in their supplementary data files. Such files should be constructed and provided in a manner considering the future reader of the paper. The content and outline of the files could easily be improved with this in mind. From my perspective Supplementary Table 1, which the authors provide in response to my question, is not necessary. Preferentially, the authors would focus their efforts on presenting their actual data in a reader accessible manner (ie where the info in Supplementary Table 1 would be available for all proteins in their dataset). If feasible, it would be preferential if a future reader could also distinguish actual measured data from imputed data in these files.

Response to point 2:

Thank you for this clarification. Please consider that it should not be the task of future readers to check such details; it is expected that the authors have done this type of evaluation. Are there other data in the paper where such information should be included (I have not checked other sites)?

Response to point 3:

I appreciate that the authors have expanded on the methods section. Yet, there are still several aspects that are only presented superficially. One being handling of multiple phosphorylated peptides. Another being details on analyses performed. Details should be such that an experiment or an analysis can be reproduced by others. Further expanding on the methods section would, in my opinion, be beneficial for future readers.

Response to point 3 part 2:

With the added methods section it is mentioned that the authors imputed 21% of the phosphoproteomics dataset. 21% is a substantial part of the dataset. Given such an extend of imputation it becomes critical that the authors systematically evaluate that their results are not driven by imputed data. This ought to be included in a global analysis.

Response to point 4:

The authors have indeed included some information on this in the revised version. As mentioned above, the authors ought to see it as their responsibility to present their data in a manner easily accessible to their future readership. I would encourage the authors to improve the outline of their supplementary data files as well as their descriptions of the file content.

Response to point 5:

The authors have indeed included information on multiplicity in the Excel file. Yet again, it has not exactly been done in a way where the information is presented in a reader-accessible format. Furthermore, the challenge of multiplicity has not really been addressed. The authors provide an example of MAPK14 phosphorylation and explain that they have data on phosphorylation levels of MAPK14 T180 from mono-phosphorylated peptides and measurements for MAPK14 T180+Y182 from doubly-phosphorylated peptides. The point is that such information is not accessible to a future reader (unless s/he has in-depth experience with MaxQuant and phosphoproteomics). In my opinion, it is the responsibility of the authors to present their data in a correct manner, and in a manner accessible to future readers. Hence, the data should be reported in a manner where the examples they gave for MAPK14 T180 and MAPK14 T180+Y182 phosphorylation can be deduced from the tables.

Response to point 5 part 2: The authors discuss IRS1_S1211 as one example in their response. I find it peculiar that the reported intensities for phosphorylation of IRS1_S1211 and IRS1_S1212 are identical in their supplementary table. Is this maybe an example of a double phosphorylated peptide (phosphorylation of S1211+S1212 but instead reported twice for two separate sites)? The biological function may be different for those two cases and accordingly the specifics of the reporting is important.

Response to point 6:

Please explain in detail how you perform this analysis when also including multiphosphorylated peptides. Do you assign the same fold-change to all sites? How do you find out which are the regulated sites (in the output file there is only 1 site listed so how do you find out which the other sites are)? If you have fold-change information for a site from a mono-phosphorylated peptide as well as from a multiphosphorylated peptide, then what do you do? Please expand on details.

Response to point 7:

In addition to statistical approaches, there is a very important consideration to be made, which is related to reproducibility. Considering the importance of reporting findings that are reproducible the response of the authors for reporting findings based on two replicates seems ill justified. Especially so as the findings are not backed by independent experiments. If the authors find it outside the scope of their efforts to repeat the entire experiment it would be a possibility to repeat a part of the experiment.

Response to point 9:

Including positive as well as negative controls is a crucial part of any experimental design. Biology behaves unexpected in many ways, and simply expecting a particular response does not come across as scientifically sound. The experiment would have been better if this control had been included from the start, but as minimum one could perform an experiment for the controls only if not the full experiment.

REVIEWER COMMENTS

Reviewer #2 (Remarks to the Author):

I have only two minor comments:

Supplementary Fig. 1c: The finding that the nuclear localization of IR is not increased after insulin exposure is apparently in contrast with the findings of Hancock et al. (Cell 177, 722–736, 2019) that showed increased chromatin-bound IR and IR interaction with transcriptional machinery in response to insulin stimulation. The hypothesis that: ‘Another potential mechanism of LYK-I effects could be direct transcriptional control by the IR or some fragments of the IR in nucleus as suggested by the work of Hancock et al (Cell 2019)’ seems also in contrast with Hancock’s findings.

We thank the reviewer for this comment. The reviewer is correct, in our data and under the conditions of these experiments, the nuclear localization of IR is not increased after insulin stimulation. However, IR and the mutated receptors are identified in extracts of the nuclear fraction in both the basal and stimulated states. We think these receptors in the nucleus in the basal acting with transcription factors and other proteins could potentially contribute to regulation of transcription, but that was not specifically addressed in these studies. We have added to a sentence on page 24 to clarify this point.

(Page 24)

“These receptors in the nucleus in the basal and stimulated states acting with transcription factors and other proteins could potentially contribute to regulation of transcription^{60,62}.”

Supplementary Fig. 8: It is unclear why the Authors chose to reconstitute the mouse IR with the IR-A isoform while they used the human IR-B isoform in all the other experiments. The legend of Fig. 8 may contain a typo: ‘(a) Immunoblot analysis of WT preadipocytes, DKO preadipocytes and DKO preadipocytes reconstituted with mouse IR (mIR, A isoform) **cells** for IR β , IGF1R β and β -actin, and quantification of IR β .’

We apologize for the unclear representation. The insulin receptor has two isoforms, A isoform and B isoform. The A isoform in which exon 11 is absent, and the B isoform in which exon 11 is included. As the reviewer is aware, inclusion of exon 11 results in the addition of 12 amino acids upstream of the intrinsic furin proteolytic cleavage site. Downstream post-translational events of either isoform result in the formation of a

proteolytically cleaved α and β subunit, which upon combination are ultimately capable of homo or hetero-dimerization to produce the disulfide-linked transmembrane insulin receptor. For reconstitution experiments, we used both A and B isoforms (the IR-A isoform for mouse IR and the IR-B isoform for human IR). The IR-A isoform showed similar LYK-I actions as the IR-B isoform. For Western blotting, we used the antibody for **IR β subunit** which detects both receptor isoforms. We have added this information in the Figure Legends to increase the clarity.

Reviewer #3 (Remarks to the Author):

I think the authors have satisfactorily met the reviewers critiques, it is certainly the case for mine.

Reviewer #4 (Remarks to the Author):

In this revision, the authors have provided answers to the questions raised. Yet, for the majority of points additional clarification is encouraged. Details pertaining the same questions as was put forward in the first revision is asked for below.

Response to point 1:

I would encourage the authors to improve the presentation of data in their supplementary data files. Such files should be constructed and provided in a manner considering the future reader of the paper. The content and outline of the files could easily be improved with this in mind.

From my perspective Supplementary Table 1, which the authors provide in response to my question, is not necessary. Preferentially, the authors would focus their efforts on presenting their actual data in a reader accessible manner (ie where the info in Supplementary Table 1 would be available for all proteins in their dataset). If feasible, it would be preferential if a future reader could also distinguish actual measured data from imputed data in these files.

We thank the reviewer for this comment. As suggested, we now have added supplement tables in what we hope will be the most readable way for both proteome and phosphoproteome. The datasheets for the phosphoproteome and the proteome contain all the original data without imputation, in addition to the data with imputation as described in the Methods section, so that the reader can test the impact of imputation and significance on their own using any parameters which they may wish to try.

We have also deposited our all mass spectrometry raw files in the publicly accessible database as we described in the “Data availability” section (All mass spectrometry raw files acquired for this study are available at ProteomeXchange Consortium via the PRIDE partner repository under the identifier ProteomeXchange: **PXD035587**).

Response to point 2:

Thank you for this clarification. Please consider that it should not be the task of future readers to check such details; it is expected that the authors have done this type of evaluation. Are there other data in the paper where such information should be included (I have not checked other sites)?

As noted above, we have provided all raw and analyzed data for both the proteome and phosphoproteome experiments in detailed supplement tables. Hopefully this will simplify analysis or reanalysis by any readers.

Response to point 3:

I appreciate that the authors have expanded on the methods section. Yet, there are still several aspects that are only presented superficially. One being handling of multiple phosphorylated peptides. Another being details on analyses performed. Details should be such that an experiment or an analysis can be reproduced by others. Further expanding on the methods section would, in my opinion, be beneficial for future readers.

Based on the comment, we added the information about the handling of multi-phosphorylated peptides in the Experimental Procedures section as follows.

In the Methods section (Page 32)

“The Maxquant output phosphoSTY table was processed using Perseus (version 1.5.2.11) software suite. Using all phosphopeptides including multi-phosphorylated peptides, we filtered for class-I phosphosites (localization probability >0.75) that have at least 50% valid values in all samples measured. This yielded 14,971 phosphosites with 20.9% missing values. The remaining missing values were imputed using a quantile regression approach for the imputation of left-censored missing data (QRILC) from the R package `imputeLCMD` with default parameters”

We now also describe each analysis in the supplemental dataset as a “README” sheet. We also added this information about this in the Experimental Procedures section as follows.

(Page 33)

“The protein sets based on kinase substrates (PhosphositePlus²⁴ and RegPhos²⁵) and phosphatases (the Dephosphorylation Database (DEPOD)⁷⁴) were tested using the Fisher exact test via the `Fisher_enrichment` function from the R package `ezlimma`⁶⁹ using default parameters. Other protein sets were analyzed using STRING⁷¹ for *Mus musculus* with default parameters. A detailed description of each analysis is in the “README” sheet in the supplemental phosphoproteome dataset.”

Response to point 3 part 2:

With the added methods section it is mentioned that the authors imputed 21% of the phosphoproteomic dataset. 21% is a substantial part of the dataset. Given such an extent of imputation it becomes critical that the authors systematically evaluate that their results are not driven by imputed data. This ought to be included in a global analysis.

We thank the reviewer for this comment. In the revised manuscript we provide more detail of our imputation strategy and the systematic evaluation results. This number of 21% imputed values is somewhat misleading. Readers must keep in mind that the same set includes samples for DKO cells, which lack both IR and IGF1R, and therefore may have very low or even undetectable values for some sites on their downstream targets. In addition, for all cell lines expressing one of the receptor constructs, there are basal and insulin stimulated values; and again the basal samples may have very low or even undetectable values for sites on their downstream targets. The number of 14,971 phosphosites with 20.9% missing values comes when we count phosphosites that have at least 50% valid values in all samples and include both the basal and insulin stimulated samples, where basal and DKO samples represent more than 50% of all samples analyzed. Therefore, we do not consider the number of “missing” or “imputed” values particularly high.

The heatmap analysis with hierarchical clustering using these data demonstrates very specific phosphorylation patterns for each IR type, as well as changes in the basal and the insulin-stimulated states with a total of 1,963 phosphosites significantly regulated among the five cell lines in either the basal or the stimulated state (Fig. 2c). When we keep phosphopeptides that have at least 80% detectable, valid values in all samples, it obviously decreases the need for imputation and decreases the number of missing values, but it also decreases the number of analyzed phosphopeptides, leaving 7,944 phosphosites with 9.1% missing/imputed values. The heatmap analysis corresponding to this approach to filtering still shows 1,704 significantly regulated phosphopeptides (new Supplementary Fig. 2b), which significantly overlaps with the phosphosite analysis in Fig. 2c ($p < 10^{-15}$ by one-sided Fisher exact test), and generates a similar heatmap to Figure 2c. We prefer the former analysis, since as noted above, more than 50% of samples come from unstimulated cells or cells with no insulin or IGF-1 receptor. Other papers in the literature (eg., PMID: 32520347 and PMID: 28369175) use a similar filtering and imputation approach with 50% valid values of samples before imputation for missing values, supporting this approach to filtering and imputation. Based on the reviewer’s comment, however, in the

revised manuscript we have now added **New Supplementary Fig. 2b** and have described this approach in more detail in the Results section on page 8 and the Experimental Procedures section on page 32 as outlined below. The information of the heatmap analysis corresponding to this filtering method has also been added in the supplemental phosphoproteomic datasheets.

(Page 8)

“It is worth noting that if we had limited our analysis to phosphopeptides that have at least 80% observed values in all samples, this would reduce the number of both the number of phosphosites and proportion of missing values by about half (7,944 phosphopeptides with 9% missing values), but the resulting heatmap would be similar in pattern to that Figure 2c (see Supplementary Fig. 2b for comparison).”

(Page 32)

“As a sensitivity analysis of our filtering paradigm, we examined the results using a paradigm in which we kept only phosphopeptides that had at least 80% observed values in all samples. This reduced both the number of analyzed phosphosites and proportion of missing values by about half (7,944 phosphopeptides with 9% missing values). The heatmap analysis corresponding to this filtering paradigm revealed 1,704 significantly regulated phosphopeptides (Supplementary Fig. 2b) which appeared similar to the Figure 2c heatmap and has a very significant overlap in the identified phosphosites ($p < 10^{-15}$ by one-sided Fisher exact test).”

Response to point 4:

The authors have indeed included some information on this in the revised version. As mentioned above, the authors ought to see it as their responsibility to present their data in a manner easily accessible to their future readership. I would encourage the authors to improve the outline of their supplementary data files as well as their descriptions of the file content.

We definitely agree with the reviewer that data should be presented in their entirety and as clearly as possible. This is not easy in an experiment with this many different cell lines and both stimulated and basal conditions, since even for us, we have spent many hours exploring this dataset. Based on the comment, however, we have worked to further improve the descriptions of the file contents as much as we can. We have also described the information in each data analyses by adding a “README” sheet in the supplemental dataset explaining the analysis, in addition to the analysis itself. Hopefully, this “README” sheet will be of help to readers who wish to access the supplemental information. With this additional information, we think our dataset has more accessible to readers. We have also added information about this in the Experimental Procedures section as follows.

(Page 33)

“A detailed description of each analysis is in the “README” sheet in the supplemental phosphoproteome dataset.”

Response to point 5:

The authors have indeed included information on multiplicity in the Excel file. Yet again, it has not exactly been done in a way where the information is presented in a reader-accessible format. Furthermore, the challenge of multiplicity has not really been addressed. The authors provide an example of MAPK14 phosphorylation and explain that they have data on phosphorylation levels of MAPK14 T180 from mono-phosphorylated peptides and measurements for MAPK14 T180+Y182 from doubly-phosphorylated peptides. The point is that such information is not accessible to a future reader (unless s/he has in-depth experience with MaxQuant and phosphoproteomics). In my opinion, it is the responsibility of the authors to present their data in a correct manner, and in a manner accessible to future readers. Hence, the data should be reported in a manner where the examples they gave for MAPK14 T180 and MAPK14 T180+Y182 phosphorylation

can be deduced from the tables.

We thank the reviewer for this comment. We have now provided the phosphoproteome data with a separate multiplicity column for each of the phosphosites in a detailed supplement table (“Multiplicity in the dataset”), in addition to our all mass spectrometry raw files in the publicly accessible database. As we described in the response to the comment 4 above, a “README” sheet has also been added to help access the supplemental information. We have included an explanation of this in the methods section ‘Dealing with multiplicity in the phosphoproteome’ providing statement on the singly, doubly, multiple phosphorylated sites.

(Page 34)

“Singly phosphorylated (multiplicity of 1): A phosphopeptide with single phosphosite quantified example MAPK14_T180_M1 represents the singly phosphorylated peptide. Multiplicity of 2 (M2), MAPK14_T180_M2 and MAPK14_Y182_M2 share the same intensity as they represent the two phosphosites on the same peptide. Multiplicity of 3 (M3), indicates phosphosites located on phosphopeptides with more than two phosphorylations. The Maxquant tool reports the intensities of a phosphosite on peptides. For example, the intensity of MAPK14_T180_M1 and MAPK14_T180_M2 differs as they come from two peptide. For each phosphosite on a multiple phosphorylated peptide, we receive a row with the same intensities as these phosphorylations are localized on the same peptide. While MAPK14_T180_M1 represents the singly phosphorylated peptide, MAPK14_T180_M2 and MAPK14_Y182_M2 share the same intensity as they represent the two phosphosites on the same peptide. Different phosphosites on the same peptide can have slightly different fold changes due to data filtering and imputation.”

Response to point 5 part 2: The authors discuss IRS1_S1211 as one example in their response. I find it peculiar that the reported intensities for phosphorylation of IRS1_S1211 and IRS1_S1212 are identical in their supplementary table. Is this maybe an example of a double phosphorylated peptide (phosphorylation of S1211+S1212 but instead reported twice for two separate sites)? The biological function may be different for those two cases and accordingly the specifics of the reporting is important.

We thank the reviewer for this comment. The Maxquant reports the phosphosites intensities derived from phosphopeptides measured. In a doubly phosphorylated peptide, the intensities for two sites are assigned the same, as they come from the same peptide

quantified. Moreover, two singly phosphorylated same peptides could also share same phosphosites intensities and would differ on their charge state. In our dataset we have both these described situation for IRS1_S1211 and IRS1_S1212 (The data are available in the supplement table). In the example for IRS1_S1211 and IRS1_S1212, the phosphosite intensity reported are from singly phosphorylated peptide (multiplicity 1), however these peptides carried charge state 2 and 3. Therefore, these sites carry same intensities but from two singly phosphorylated peptides. Detailed information on this point is now present in the “Multiplicity in the dataset” table in the supplement datasheets.

Response to point 6:

Please explain in detail how you perform this analysis when also including multiphosphorylated peptides. Do you assign the same fold-change to all sites? How do you find out which are the regulated sites (in the output file there is only 1 site listed so how do you find out which the other sites are)? If you have fold-change information for a site from a mono-phosphorylated peptide as well as from a multiphosphorylated peptide, then what do you do? Please expand on details.

All phosphopeptides quantified, including multiphosphorylated peptides, were used for the analysis. Phosphosites derived from multi-phosphorylated peptides that have less than 50% valid values of all samples are excluded from the analysis. This is described in the Experimental Procedures section.

We report the phosphosite intensities of all the quantified and regulated sites, independent of the phosphosite multiplicity, since the biological function may be different based on its multicity form. In the case of representing selected known (e.g., IRS1_S1211) or novel phosphosite from the dataset, if two of the same phosphosites with different multiplicity were found to be significantly regulated, we choose the singly phosphorylated peptide to represent the data, to help keep the interpretation straight forward. In this case, IRS1_S1211 was detected as a singly phosphorylated peptide and as a doubly phosphorylated peptide, with the direction of regulation being “up” in both the cases independent of its multiplicity.

Response to point 7:

In addition to statistical approaches, there is a very important consideration to be made, which is related to reproducibility. Considering the importance of reporting findings that are reproducible the response of the authors for reporting findings based on two replicates seems ill justified. Especially so as the findings are not backed by independent

experiments. If the authors find it outside the scope of their efforts to repeat the entire experiment it would be a possibility to repeat a part of the experiment.

As described in the manuscript, the aim of this study was to determine the roles of IR which might be ligand and tyrosine kinase-independent (LYK-I). The data clearly show changes in phosphorylation, gene/protein expression and biological function indicating a class of IR actions which are dependent on the intracellular domain, but independent of ligand binding or tyrosine kinase activity as summarized schematically in Figure 8e. Based on the biological meaning and importance, most important part of this study is DKO, IR (wild-type receptor) and K1030R-IR (kinase-dead receptor) cells at the basal state (no stimulation by insulin). While for technical reasons, one of the triplicate samples was lost in the phosphoproteome analysis for Δ CT_basal, Δ CT_stim and JMO_stim in the phosphoproteome data, all other data, include the WT IR and kinase dead IR were studied with an N=3 or more. In the phosphoproteome analysis, we focused mainly on these samples which defined the “High in IR-ICD” and the “Low in IR-ICD” clusters, and which were associated with changes in gene/protein expression indicate an IR signaling pathway which is dependent on the intracellular domain, but ligand and tyrosine kinase-independent (LYK-I). In terms of the phosphoproteomics, these phosphorylations were clearly up- or down-regulated in IR and K1030R compared to DKO cells with insulin-independent manner with n=3 for basal and stimulated states. In these clusters, all four Δ CT samples (2 basal and 2 insulin-stimulated) also clearly showed the same pattern indicating these were insulin-independent, and these changes were not observed in JMO basal samples (N=3) indicating the dependence of the IR intracellular domain. We think these findings in the heatmap analysis with statistically significant show very clear signaling modifications from ligand and tyrosine kinase-independent IR.

The Akt-phosphorylation differences related Figure 1d and 2e were also confirmed in triplicate by Western blotting (new supplemental figure 2c). With literally hundreds of changes, it would be impractical to confirm the majority of them by western blotting, especially since for many of these sites no validated antibodies exist.

(Page 9)

“The phosphorylation of Akt3^{S472} as assessed by phosphoproteomics gave similar results to the western blot data of Akt phosphorylation in Figure 1d and Supplemental Figure 2c, in which the phospho-Akt antibody used can recognize Akt1 (S473), Akt2 (S474) and Akt3 (S472).”

Response to point 9:

Including positive as well as negative controls is a crucial part of any experimental design. Biology behaves unexpected in many ways, and simply expecting a particular response does not come across as scientifically sound. The experiment would have been better if this control had been included from the start, but as minimum one could perform an experiment for the controls only if not the full experiment.

The aim of this study was to determine the potential roles of IR which are ligand and tyrosine kinase-independent (LYK-I). Therefore, our paper has mainly focused on the basal changes (non-stimulated by insulin). Western blotting data of the new Supplementary Fig. 8b showed no stimulation of phospho-IRS1 and phospho-Akt in DKO cells by insulin, which are part of the classical core downstream insulin signaling pathway. These confirm that these cells are truly DKO cells and lack these core insulin stimulated downstream signals. We did not assess DKO cells stimulated by insulin since this seems an unnecessary expense and work. In any case, this does not affect the results of changes in basal phosphorylation between DKO, IR and IR mutants which is main aim of this study. Together, these data reveal a group of ligand and tyrosine kinase independent (LYK-I) actions of the IR, which can be visualized at the level of the phosphoproteome, but also has effects at the level of mRNA expression, the proteome, and in cell function.

REVIEWERS' COMMENTS

Reviewer #2 (Remarks to the Author):

The Authors have answered my questions.

I still believe that the following sentence in legend to supplemental Fig. 8: '(a) Immunoblot analysis of WT preadipocytes, DKO preadipocytes and DKO preadipocytes reconstituted with mouse IR (mIR, A isoform) cells for IR-beta subunit (IR β), IGF1R-beta subunit (IGF1R β) and β -actin, and quantification of IR β . Data are means \pm SEM (n = 3 per group). *** P < 0.001. ' should be: (a) Immunoblot analysis of WT preadipocytes, DKO preadipocytes and DKO preadipocytes reconstituted with mouse IR (mIR, A isoform) for IR-beta subunit (IR β), IGF1R-beta subunit (IGF1R β) and β -actin, and quantification of IR β . Data are means \pm SEM (n = 3 per group). *** P < 0.001.

Reviewer #4 (Remarks to the Author):

The authors have addressed my points of critiques satisfactorily.